# Altered Transcriptional Regulation of Glycolysis in Circulating CD8^+^ T Cells of Rheumatoid Arthritis Patients

**DOI:** 10.3390/genes13071216

**Published:** 2022-07-07

**Authors:** Shilpa Harshan, Poulami Dey, Srivatsan Raghunathan

**Affiliations:** 1Institute of Bioinformatics and Applied Biotechnology, Bangalore 560100, Karnataka, India; hshilpa@ibab.in; 2Manipal Academy of Higher Education, Manipal 576104, Karnataka, India; 3The Edu Bio LLC, 2222 W. Grand River Ave, STE A, Okemos, MI 48864, USA; poulami2912@gmail.com

**Keywords:** rheumatoid arthritis, T lymphocyte, regulation of glycolysis, RNA sequencing, *PFKFB3*, pentose phosphate pathway, *GAPDH*, *MYC*, *HIF1A*, mTOR

## Abstract

Peripheral T lymphocytes of rheumatoid arthritis (RA) patients show pathological changes in their metabolic pathways, especially glycolysis. These changes may drive the increased proliferation and tissue invasiveness of RA T cells. In order to study the transcriptional regulation underlying these alterations, we analysed publicly available RNA sequencing data from circulating T lymphocyte subsets of healthy individuals, untreated RA patients, and patients undergoing treatment for RA. Differential co-expression networks were created using sample-wise edge weights from an analysis called “linear interpolation to obtain network estimates for single sample” (lionessR), and annotated using the Gene Transcription Regulation Database (GTRD). Genes with high centrality scores were identified. CD8^+^ effector memory cells (Tem) and CD8^+^CD45RA^+^ effector memory cells (Temra) showed large changes in the transcriptional regulation of glycolysis in untreated RA. *PFKFB3* and *GAPDH* were differentially regulated and had high centrality scores in CD8^+^ Tem cells. *PFKFB3* downregulation may be due to *HIF1A* post transcriptional inhibition. Tocilizumab treatment partially reversed the RA-associated differential expression of several metabolic and regulatory genes. *MYC* was upregulated and had high centrality scores in RA CD8^+^ Temra cells; however, its glycolysis targets were unaltered. The upregulation of the PI3K-AKT and mTOR pathways may explain *MYC* upregulation.

## 1. Introduction

T lymphocytes are major players in the inflammatory autoimmune disease rheumatoid arthritis (RA). RA is characterised by painful inflammation of usually small joints, where the synovial tissue lining the joints becomes hyperplastic and invades the underlying cartilage. Patients with RA show manifestations in the blood several years before there are any symptoms in the joints [1]. This is known as the pre-clinical stage of RA.

The shift from pre-clinical stage with circulating auto-antibodies to clinical synovitis is marked by changes in T cell populations in the blood. The reduction in naive CD4^+^ T cells and regulatory T cells (T reg) and the appearance of an abnormal T cells subset named inflammation-related cells (IRC) are predictive of progression to synovial [2]. A defective differentiation process of naive CD4^+^ T cells results in the production of short-lived effector cells (SLEC) instead of long-lived memory precursor cells. The SLEC are highly proliferative and invade the synovial tissue, causing synovitis. This pathological differentiation is thought to be caused by defects in DNA repair and cellular metabolism in the naive CD4^+^ T cells of susceptible patients [3]. While effector cell populations increase, regulatory T cell populations decrease in the peripheral blood [4,5]. They also become less effective.

The defects in T cell DNA repair mechanisms result in reduced telomere stability and the accumulation of errors in mitochondrial DNA. Damaged mitochondria lead to several changes in the metabolism of circulating RA T cells. In healthy individuals, naive T cells (Tn) primarily depend on mitochondrial fatty acid (FA) oxidation for energy. Upon activation, T cells need to rapidly proliferate. This sets up a large demand for energy and substrates for DNA synthesis, protein synthesis and membrane formation. In order to satisfy this demand, activated T cells shift their main energy source to glycolysis and oxidative phosphorylation. Even though glycolysis produces less ATP than oxidative phosphorylation, glycolysis and linked pathways such as the tricarboxylic acid (TCA) cycle and the pentose phosphate pathway (PPP) produce metabolites to be used as precursors in nucleic acid synthesis. Activated cells differentiate into various forms of effector cells based on extracellular signals and the availability of metabolic substrates. The cells eventually die or turn into memory cells, which are no longer proliferative, and use FA oxidation for [6]. FA oxidation, TCA cycle and oxidative phosphorylation are all mitochondrial pathways. It is evident that mitochondria are essential for metabolism during every stage of T cell differentiation. Thus, the effect of mitochondrial damage on T cell function is profound.

In the place of glycolysis and oxidative phosphorylation, circulating RA T cells use the pentose phosphate pathway. This leads to reduced pyruvate production and an increased production of NADPH. This, combined with reduced reactive oxygen species (ROS) production due to impaired oxidative phosphorylation, sets up a reducing environment within the T cell. ROS are required for intracellular signalling, including for the activation of the repair enzyme Ataxia Telangiectasia Mutated (ATM). These changes to signalling create a pro-inflammatory phenotype for the T cell even before they enter the synovium [7].

The other change associated with reduced mitochondrial function is linked to the TCA cycle and FA oxidation, both of which occur in the mitochondria. TCA cycle utilises acetyl CoA. In the context of reduced TCA cycle, the acetyl CoA is diverted to fatty acid synthesis that occurs in the cytosol. At the same time, mitochondrial FA oxidation is reduced, and NADPH required for FA synthesis is generated by the upregulated pentose phosphate pathway. These three events give the cell an increased FA presence, which accumulates as lipid droplets. The excess lipids allow the cells to form extensive membrane structures, which enable them to invade tissues more easily. Thus, the disruption of mitochondrial function contributes to both the proliferative and tissue-invasive nature of SLECs present in the blood of RA [7].

These changes were demonstrated in CD4^+^ T cells in RA. CD8^+^ T cells appear to have a different program of metabolic regulation. Circulating CD8^+^ T cells in RA appear to have higher levels of aerobic glycolysis and lactate production than their healthy counterparts [8]. In contrast to CD4^+^ T cells, CD8^+^ T cells in RA had higher ROS production than healthy CD8^+^ T cells. The CD8^+^ T cells in RA peripheral blood were also more proliferative than their healthy counterparts and are able to induce a pro-inflammatory phenotype in B [8].

Therefore, it is evident that the aberrant metabolic program of CD4^+^ T cells contribute to the pathogenesis of RA. A similar relation between metabolism and increased synovitis may also exist for CD8^+^ T cells. This makes metabolism and the regulation of metabolism promising targets for therapy in RA.

The regulation underlying these altered metabolic patterns in CD4^+^ T cells is thought to be mediated by an increased ratio between the enzymes 6-Phosphofructo-2-Kinase/Fructose-2,6-Bisphosphatase 3 (PFKFB3) and glucose-6-phosphate-dehydrogenase (G6PD). PFKFB3 catalyses the synthesis of fructose 2,6 bisphosphate, an allosteric modulator of the glycolysis enzyme phosphofructokinase 1 (PFK1), while G6PD catalyses the first-rate limiting step of the pentose phosphate pathway, converting glucose 6-phosphate to 6-phosphogluconolactone. A decrease in the ratio between PFKFB3 and G6PD pushes the cells to the pentose phosphate pathway. In addition, the mechanistic target of the rapamycin complex 1 (mTORC1) pathway is activated due to reduced 5′ adenosine monophosphate-activated protein kinase (AMPK) signalling, resulting in increased anabolism. The roles of hypoxia-inducible factor 1-α (HIF1A) and MYC, which are responsible for the regulation of aerobic glycolysis in healthy T cells are not clear [3].

CD8^+^ T cells also showed increased mTORC1 activation due to reduced levels of the inhibitor Tuberous Sclerosis Complex 2 (TSC2) and PRKAA1(alpha1 sub unit of AMPK). They also have higher levels of the *HIF1A* mRNA as measured by a microarray, which is thought to be responsible for the increased expression of several glycolytic enzymes [8].

Although the metabolic pathways that affect circulating T cell function in RA have been explored, the transcriptional regulation of these genes are less understood. Since these cells show pathological changes outside of the affected synovial compartment, examination of the transcriptome of T cells isolated from the blood of RA patients may provide insights into the transcriptional regulation associated with metabolic changes. In order to do this, we planned to study any RNA sequencing data in peripheral blood T cells of RA patients submitted to open-source repositories. We studied the publicly available RNA sequencing data set GSE118829 from the Sequence Read Archive (SRA) [9]. The data set consisted of samples from seven T cell subsets of healthy, untreated and treated RA individuals. The details of the samples and T cell subsets are given in the methods section. Since glycolysis appears to be a major affected pathway in both CD4^+^ and CD8^+^ T cells, we decided to focus on the regulation of enzymes related to glycolysis. We identified differentially expressed genes (DEGs) in the glycolysis pathway and other pathways related to energy metabolism in the different T cell subsets of untreated RA individuals in comparison with healthy individuals. For the purpose of examining the relationship of these genes with transcription factors (TF), we annotated transcription factors in the data set using the database Gene Transcription Regulation Database (GTRD) [10] and glycolysis-related genes using the Kyoto Encyclopedia of Genes and Genomes (KEGG) pathway “Glycolysis/gluconeogenesis” [11]. We supplemented the KEGG pathway annotation with a literature search to identify transporters, enzymes controlling the concentration allosteric modifiers, post transcriptional modifiers and enzymes that affect the concentration of glycolysis metabolites. We finally created a list of genes which we called the “TF-glycolysis gene list”. We calculated single sample edge weights for gene pairs of this list using the R/Bioconductor package “lionessR” (linear interpolation to obtain network estimates for single samples), based on the correlation between their counts [12]. This method is described in detail in the Methods section.

For each T cell subset, we then identified those gene pairs that showed a significant difference in edge weights between the RA and healthy samples by performing a statistical test using a linear model in the “limma” package of R/Bioconductor [13]. Among the differentially co-expressed gene pairs, we retained those that were annotated by the GTRD database. We constructed a network of these genes for each T cell type and then performed a network analysis. In these networks, the nodes are genes, and the edges connect pairs of differentially co-expressed genes that are annotated by GTRD.

In each network, we finally chose those nodes that are common among the top ten nodes in each of the three centrality measures considered. The centrality measures and selection criteria are described in the Methods section. The DEGs were identified in these networks, and in the major metabolic pathways.

The target genes with high centrality measures can be considered as those genes whose transcriptional regulation changed in the RA condition. The TF neighbours of these genes are thus likely to be responsible for driving any RA-related change in the expression of glycolysis-related genes. Similarly, transcription factors with high centrality scores that also have links to important glycolysis-related genes may be central to these RA-related changes to the regulation of glycolysis.

Since the activity of metabolic pathways depends on several post transcriptional events such as localization, post-translational modifications, allosteric regulation, substrate availability, etc., this study was not able to measure these changes. However, to date, this is the most comprehensive study on the regulation of glycolysis in RA peripheral T lymphocytes at the level of transcription.

Our study shows that CD8^+^ T effector memory (Tem) cells and CD8^+^ CD45RA^+^ effector memory (Temra) cells have large numbers of DEGs. These cell types also show large numbers of significantly different gene pairs annotated by GTRD.

In the untreated RA CD8^+^ Tem cells, the enzymes catalysing the rate-limiting steps in glycolysis are not differentially expressed at level of transcription. Among the regulators of glycolysis, *PFKFB3*, is downregulated and has high target centrality scores in the GTRD-annotated differential co-expression network reported in this study. The downregulation of PFKFB3 is known to shift the CD8^+^ Tem from glycolysis to PPP. We propose a mechanism for the downregulation of PFKFB3 by the post transcriptional inhibition of HIF1A. One of the upregulated inhibitors of HIF1A, glyceraldehyde-3-phosphate dehydrogenase (*GAPDH*) also shows high target centrality scores in the network. The connection of *GAPDH* to multiple differentially expressed TFs in our network potentially indicates a wide range of functions beyond glycolysis for this gene in RA CD8^+^ Tem cells.

Our study shows that tocilizumab (TCZ) treatment partially reverses the RA associated differential expression of several metabolic and regulatory genes. This includes genes encoding mitochondrial complex I proteins, ATP synthase sub units and TCA cycle enzyme sub units in CD8^+^ Tem.

In untreated RA CD8^+^ Temra cells, the transcription factor *MYC* is upregulated and has a high centrality score in our network. However, the transcriptional target genes of *MYC* in glycolysis are not upregulated in this study. *MYC* upregulation is possibly caused by signalling via the phosphatidylinositol 3′-kinase (PI3K)-AKT and the mechanistic target of rapamycin (mTOR) pathways. Here, we report the upregulation of the activators of these pathways.

To summarize, RA causes changes to the transcriptional regulation of glycolysis in CD8^+^ Tem and Temra cells. We propose possible mechanisms for these changes. The RA associated alterations are partially reversed by the TCZ treatment.

## 2. Materials and Methods

The workflow used in this project is shown in Figure 1. RNA sequencing data of T cell subsets from healthy individuals, individuals with untreated RA and individuals treated with one of three therapies were subjected to differential expression analysis. The co-expression analysis of the transcription factors and glycolysis-related genes were performed using lionessR. The differentially co-expressed gene pairs were identified using limma and annotated with transcription factor–target gene interactions using the GTRD database. A network of annotated gene pairs was created, and network analysis was performed. From this analysis, the transcription factors and target genes playing important roles in changing the regulation of glycolysis under RA conditions were identified. The details of each step are described below.

### 2.1. RNA Sequencing Data

The data set GSE118829 was downloaded from the SRA database using the SRAtoolkit command fastq-dump [14]. A total of 336 samples were present in the data set, from the blood or synovial fluid of the following six groups:Untreated RA (T cell subsets from blood);Healthy control (T cell subsets from blood);Infliximab and methotrexate combination therapy (T cell subsets from blood);Methotrexate monotherapy (T cell subsets from blood);Tocilizumab monotherapy (T cell subsets from blood);RA synovial fluid samples (T cell subsets).

Infliximab and methotrexate combination therapy is referred to as infliximab-treated or IFX for the remainder of this article.

Each of the above six cohorts had samples for CD4^+^ and CD8^+^ T cell subsets defined using the following markers:T naive (Tn): CD45RA^+^ CCR7^+^;T central memory (Tcm): CD45RA^−^ CCR7^+^;T effector memory (Tem): CD45RA^−^ CCR7^−^;T effector memory CD45RA^+^ (Temra): CD45RA^+^ CCR7^−^.

The Temra samples contained only CD8^+^T cells, while Tn, Tcm and Tem samples had both CD4^+^ and CD8^+^ T cells.

A quality control (QC) was performed on the fastq files using R/Bioconductor libraries. The reads were mapped to GRCh38.p13 using STAR 2.7.7a [15]. Of the 336 single-end fastq files downloaded, 317 files passed the QC test with greater than 70% reads mapped to the genome. The details of each sample group are given in Table 1. The synovial fluid samples were not used in any of the analyses due to their small sample size.

### 2.2. Differential Expression Analysis

The aligned reads were counted using the R/Bioconductor package Rsubread (version 2.4.3) [16]. Differential expression analysis was performed using the DESeq2 (version 1.36.0) [17]. Genes with an FDR corrected *p* value ≤ 0.1 and a linear fold change of 1.5 were considered to be differentially expressed. The following comparisons between the sample groups were made for each T cell subset:Untreated RA vs. healthy control;Infliximab/methotrexate (IFX) treated vs. untreated RA;Methotrexate (MTX) treated vs. untreated RA;Tocilizumab (TCZ) treated vs. untreated RA.The KEGG Mapper “color” tool was used to map the DEGs to KEGG [18].

### 2.3. Identification of Transcription Factors and Glycolysis-Related Genes

In order to study the differential co-expression of glycolysis-related genes and their transcription factors, we proceeded as follows. Among the genes in the RNA sequencing data, we identified those that were also present in the KEGG pathway “Glycolysis/gluconeogenesis”. Furthermore, a literature search was performed to identify any transporters or enzymes that directly affect the glycolysis pathway enzymes by post translational modification or changing the concentration of key intermediates in the glycolysis pathway.Transcription factors were annotated using the database GTRD (v19). We finally created a list of genes which we called the “TF-glycolysis gene list”. This list included 1233 transcription factors and 74 glycolysis-related genes.

### 2.4. Single Sample Edge Weight Calculation

The raw counts generated from the aligned BAM files were transformed using the DESeq2 package function rlog before this analysis. From this counts matrix, genes belonging to the TF-glycolysis gene list were selected. The following analysis were performed on this subset matrix.The R/Bioconductor package lionessR version (1.10.0) was used to calculate single-sample edge weights for each gene pair among the TF-glycolysis gene list. This package uses the method known as Linear Interpolation to Obtain Network Estimates for Single Samples (LIONESS).The correlation matrix of gene expressions across samples is computed in lionessR.Each element of the matrix represents the correlation coefficient between expressions of pairs of genes across all the samples. By computing this correlation matrix with and without a particular sample, an “edge weight” is assigned, which is an estimate of the contribution of that sample to the overall co expression between pairs of genes. Thus, every possible gene pair in the TF-glycolysis gene list has an edge weight in each sample.

### 2.5. Differential Edge Weight Calculation

In order to determine which gene pairs had a significantly different edge weight between the compared categories, a differential edge weight analysis was performed using limma package of Bioconductor (version 3.52.1).The following groups of samples were compared with each other for each T cell subset:Untreated RA vs. healthy control;Infliximab/methotrexate (IFX) treated vs. untreated RA;Methotrexate (MTX) treated vs. untreated RA;Tocilizumab (TCZ) treated vs. untreated RA.The significantly different gene pairs were chosen using an FDR-adjusted *p* value cutoff of ≤0.1.

### 2.6. Creation of GTRD-Annotated Differential Co-Expression Networks

The GTRD database contains transcription factor target gene interactions based on the presence of at least one binding site for a transcription factor in the region −1000 to +100 base pairs from the transcription start site of a given gene. GTRD does not contain data for lymphocytes from RA patients or for CD8^+^ Temra cells. Therefore, we decided to use the integrated data provided by GTRD, which included data from all human cell types present in the database [10]. The transcription factor—target gene interactions can be considered as a directed network with the transcription factor being the source node and the target gene being the target node. We used this information to annotate the significant gene pairs from the differential analysis of the edge weights from lionessR. Significant gene pairs which were also present in the GTRD database as transcription factor—target gene pairs were selected, and the direction assigned. This resulted in a network where an edge was present between two genes if:The edge weight was significantly different between the two categories, and;The edge was present in the GTRD database.Such networks were constructed for every comparison made in each T cell subset using the R package “igraph” (version 1.3.1) [19]. This network is described in Figure 2.

The networks were visualised using Cytoscape (3.8.2) [20]. In the untreated RA vs. healthy control comparison of the edge weights, a positive log fold change value in the limma results table indicates higher edge weight in the untreated RA samples and is depicted as a purple edge in the Cytoscape network. A negative log fold change indicates higher edge weight in the healthy control samples and is indicated by a green edge in the Cytoscape network. In the treated RA vs. untreated RA samples, a positive log fold change shows a higher edge weight in the treated sample, indicated by a purple edge, and a negative log fold change shows a higher edge weight in the untreated RA sample, indicated by a green edge.

### 2.7. Network Analysis

The networks created had two types of nodes: transcription factors and target genes. None of the glycolysis-related genes chosen for this analysis were annotated as TFs in GTRD. On the other hand, any TF can also be a target gene, if another TF binds to its promoter region (−1000 to +100 from transcription start site). Therefore, in these networks, all glycolysis-related genes were target genes and only had incoming edges, while the TFs had both incoming and outgoing edges.The networks with fewer than 20 glycolysis-related genes were not considered further.Centrality measures were calculated for the nodes in each network using “igraph” package [19].The following centrality measures were calculated for each network: in-degree centrality, out-degree centrality, betweenness centrality, hub score and authority score.

#### 2.7.1. Degree Centrality

The degree of a node is the number of links associated with it. In a directed network, degree can be defined based on the direction of the edge. In-degree refers to the number of edges directed to a node, while out-degree refers to the number of edges directed out of a node [21].

#### 2.7.2. Betweenness Centrality

This is a measure of how important a node is to the connectivity of the network. Betweenness centrality is the number of shortest paths that pass through a node, where the shortest path is defined as the shortest sequence of nodes and edges that connect two given nodes [22]. In our network, target genes do not have outgoing edges and cannot be part of the shortest path between any pair of genes. Therefore, their betweenness centrality will be zero. Thus, we consider betweenness centrality for transcription factors alone.

#### 2.7.3. Hub and Authority Scores

Hub and authority scores are only defined in directed networks. Hub score depends on outgoing edges, and describes the outgoing connectivity of a node to other nodes with a large number of incoming edges. Authority score depends on incoming edges and describes the incoming connectivity of a node with other nodes that have a large number of outgoing edges. That is, nodes with high hub scores are those that have outgoing edges to a large number of high authority score nodes, while nodes with high authority scores are those that have incoming edges from a large number of high hub score nodes.Hub and authority scores of the nodes in a network are defined as relative to each other. The authority score of a node is the sum of the hub scores of the nodes that link into it. The hub score of a node is the sum of authority scores of the nodes that it links out to. These measures are calculated by first assigning a uniform hub and authority score to all nodes and iteratively updating the scores [23].Since only transcription factors have both outgoing and incoming edges, centrality measures that depend on outgoing edges, such as out-degree centrality, hub score and betweenness centrality were only considered for the transcription factors.We then ranked the transcription factors by out-degree, hub score and betweenness centrality and the glycolysis-related genes by in-degree and authority score. The top ten genes in each centrality measure were taken. The common transcription factors across the out-degree, hub score and betweenness score lists and the common glycolysis-related genes across the in-degree and authority score lists were identified.These genes were considered to be important in the difference in the TF–target gene interactions between the two compared groups, and thus central to the altered regulation of glycolysis between the two groups.

### 2.8. Annotation with GeneHancer Database

The expression of a gene may be influenced by the binding of transcription factors to enhancer regions. These regions may be several kilobases away from the location of a target gene [24]. Thus, the GTRD data that cover the −1000 to +100 region around the transcription start site may not adequately account for all the transcription factor–target interactions that are involved in the regulation of the expression of a given target gene. In order to account for transcription factor binding to enhancer regions, we annotated the differentially co-expressed edges between transcription factors and glycolysis target genes with information from the GeneHancer database (version 5.10).

GeneHancer links enhancers spread across the genome to target genes. It also provides information on the transcription factors that bind to the enhancers [25]. This allows us to create a transcription factor–target gene network using the number of enhancer binding sites of a given target gene for any given transcription factor. This network is used to annotate the differentially co-expressed edges. Since this network does not contain edges between transcription factors, the network analysis methods used for analysing the GTRD-annotated networks cannot be used. However, the in-degree of the glycolysis target genes remains comparable, since these only have incoming edges from transcription factors in both the networks. Therefore, we compared the in-degree of the glycolysis genes in GTRD-annotated and GeneHancer-annotated differential co-expression networks of untreated RA CD8^+^ Tem and Temra cells. We also compared the number of transcription factor binding sites in GTRD and GeneHancer for the selected target genes.

### 2.9. Color Pathway Tool of KEGG Mapper

The signalling pathways mTOR signalling and PI3K-AKT signalling are important regulators of glycolysis [26,27]. The mTORC1 pathway, which is part of the KEGG mTOR signalling pathway, was also shown to be upregulated in the GSE118829 data set [9]. The KEGG Mapper “color” tool was used to map the differentially expressed genes of RA CD8^+^ Tem and RA CD8^+^ Temra cells to the KEGG pathways mTOR signalling and PI3K-AKT signalling [18].

R scripts used for the analyses in this study are available at https://github.com/hshilpa/lionessR_RA_Tcell, accessed on 1 July 2022.

## 3. Results

### 3.1. Differential Expression of Genes

The number of DEGs between untreated RA and healthy individuals for T cell subsets are shown in Table 2. Genes with an FDR of less than 0.1 and a linear fold change greater than 1.5 were considered to be differentially regulated.

CD8^+^ Tem and CD8^+^ Temra cells show the largest number of differentially expressed genes.

Untreated RA samples were also compared with samples from individuals undergoing methotrexate, infliximab or tocilizumab therapy. The number of genes differentially expressed when comparing the treated and untreated RA samples is shown in Table 3.

CD8^+^ Temra cells from methotrexate (MTX)-treated individuals had the highest number of downregulated genes, followed by CD8^+^ naive cells. Similarly, IFX-treated individuals also had the highest number of downregulated genes in CD8^+^ Temra cells, followed by CD8^+^ Tn cells. TCZ-treated individuals showed a different pattern, with CD8^+^ Tem cells showing the largest number of downregulated genes. CD8^+^ Tn cells of the TCZ-treated individuals also showed a large number of downregulated genes. Overall, the number of upregulated genes was lower than or equal to that of the downregulated genes. The only exceptions were CD8^+^ Tem cells of MTX-treated individuals and CD4^+^ Tn cells of IFX-treated individuals.

### 3.2. Transcriptional Regulators Enriched in the Differentially Expressed Gene Lists

Differentially expressed genes from untreated RA CD8^+^ Tem and CD8^+^ Temra cells were analysed for enriched transcriptional regulators using Enrichr [28]. The gene set library used for performing enrichment was the consensus target genes for transcription factors from ENCODE and ChEA experiments. The upregulated gene list of CD8^+^ Tem and CD8^+^ Temra cells and the combined up and downregulated gene list of CD8^+^ Tem cells did not show significant enrichment for any transcription factor. The combined the up- and downregulated gene list of CD8^+^ Temra cells that showed *RFX5* as significantly enriched.

The enriched transcription factors in the downregulated gene list for CD8^+^ Tem cells are shown in Table 4.

The enriched transcription factors in the downregulated gene list for CD8^+^ Temra cells are shown in Table 5.

*CREB1*, *UBTF*, *ZNF384* and *ATF2* were enriched in the downregulated lists of both CD8^+^ Tem and CD8^+^ Temra cells of untreated RA cells. CREB1 is involved in T cell cytotoxicity [29]. ATF2 is involved in the response of T cells to steroid therapy [30]. The roles of UBTF and ZNF384 in the RA T lymphocytes are not known.

### 3.3. Differential Regulation of Glycolysis/Gluconeogenesis Pathway Enzymes

Genes coding for enzymes in the glycolysis pathway showed differential regulation between untreated RA and healthy samples in two cell types, CD8^+^ Tem and CD8^+^ Temra cells. The genes showing differential expression are shown in Table 6.

Although *HKDC1* and *HK2* from CD8^+^ Temra cells showed significant differential regulation, their mean counts in the untreated RA samples and the healthy samples were low. GAPDH is known to be over-expressed in the hypoxic synovial tissue of inflammatory arthritis patients, along with PKM2 and SLC2A1 [31]. However, *PKM2* and *SLC2A1* mRNA levels did not change in any CD8^+^ T cell subsets in this study. To the best of our knowledge, the levels of GAPDH in circulating T cell subsets of RA patients have not been reported. *LDHA* mRNA was shown to be increased in the CD8^+^ T cell subsets of RA patients [8]. However, *LDHA* was not upregulated in CD8^+^ T cell subsets in this study. Since this study only measured mRNA levels, we cannot comment on the protein levels or activity of LDHA in these cells.

Glycolysis/gluconeogenesis pathway enzymes were differentially expressed in some of the treated T cell subtypes. Table 7 shows these genes.

The CD8^+^ Temra cells from MTX-treated samples showed a significant downregulation of *HK2*. However, the mean counts in the untreated RA and MTX-treated samples were very low.

### 3.4. Differential Regulation of Enzymes in Other Metabolic Pathways

#### 3.4.1. Pentose Phosphate Pathway

Transaldolase 1 (*TALDO1*) and gluconokinase (*IDNK*) were the only genes to show differential regulation between RA and healthy samples. They were upregulated in CD8^+^ Tem cells from healthy samples to untreated RA samples.

*IDNK* was downregulated in TCZ-treated CD8^+^ Tem cells.

#### 3.4.2. Tricarboxylic Acid Cycle

Among untreated RA versus healthy samples, CD8^+^ Tem cells and CD8^+^ Temra cells showed a differential regulation of genes involved in the TCA cycle. Table 8 lists the differentially expressed genes.

CD4^+^ T cells from the peripheral blood of RA patients are known to have reduced levels of DLD and SUCLG2 [32]. Their levels in CD8^+^ T cells have not been reported. Although *DLD* is downregulated in both CD8^+^ Tem and CD8^+^ Temra cells, the levels of *SUCLG2* are unaltered (Appendix A).

#### 3.4.3. Oxidative Phosphorylation

Untreated RA CD8^+^ Tem cells showed a differential expression of 11 oxidative phosphorylation pathway genes. These are shown in Table 9. RA CD8^+^ T cells are known to have reduced ATP production and increased ROS production [8]. The upregulation of complex 1 genes and downregulation of the ATP synthase catalytic subunit *ATP5F1C* are concordant with these observations.

Genes from this pathway were downregulated in CD8^+^ Tem and CD8^+^ Tn cells from TCZ-treated individuals. Those genes are listed in Table 10.

#### 3.4.4. Fatty Acid Metabolism

One gene in CD8^+^ Tem cells and two genes in CD8^+^ Temra cells were differentially regulated in untreated RA samples compared to healthy samples. These are listed in Table 11.

The differentially regulated genes of IFX-treated CD8^+^ Temra cells, TCZ-treated CD8^+^ Tem cells and TCZ-treated CD8^+^ Temra cells are shown in Table 12.

*ACOT1*, which is upregulated in CD8^+^ Temra cells of untreated RA individuals, is downregulated in IFX- and TCZ-treated CD8^+^ Temra cells.

### 3.5. Sample Specific Edge Weights

The counts of genes present in the TF-glycolysis gene list were extracted and single-sample edge weights were calculated for every possible gene pair using the R/Bioconductor package “lionessR”. Differential analysis was performed on these edge weights using limma and significantly different edges were identified using an FDR-corrected *p* value threshold of 0.1. The GTRD database contains directed transcription factor target interactions. Those significant edges from each comparison that were present in GTRD were then identified. Table 13 shows the number of significant gene pairs for each cell type.

CD8^+^ Temra cells from untreated RA samples, when compared to those from healthy samples, had the highest number of significantly different edge weights. Out of 128,318 significant edges, 75,606 were present in GTRD. Among the treated samples, TCZ-treated samples had more than 100 significantly different edge weights in all cell types except in CD8^+^ Tcm cells. IFX-treated samples had a similarly large number of significantly differentially expressed genes in cell types other than CD4^+^ Tcm cells. For MTX-treated samples, CD4^+^ Tn and CD4^+^ Tcm cells had few (less than 50) differentially expressed edges, while other cell types had comparatively higher numbers of differentially co-expressed edges.

### 3.6. Network Analysis

The GTRD-annotated transcription factor–target gene edges identified as significant were used to create a directed unweighted network using the R package “igraph”. In this network, the presence of an edge indicates that: (a) there is a significant difference in the mean edge weights between the two categories being compared and (b) the interaction is present in the GTRD database. The direction of the edge is taken from GTRD, and there are no edge weights. Table 14 shows the total number of nodes and edges as well as the number of transcription factors and target genes present in each network.

When comparing untreated RA samples with healthy samples, CD4^+^ Tn cells and CD4^+^ Tcm cells had four and zero glycolysis-related genes in their networks. Among the comparisons of the treatment samples with the untreated samples, CD4^+^ Tcm cells and CD8^+^ Tcm cells had fewer than 20 glycolysis-related genes in the network. CD4^+^ Tn cells also had only eight and zero glycolysis-related genes in the TCZ- and MTX-treated samples. Since these networks had low numbers of glycolysis-related genes with significantly altered edge weights, we did not consider them for further network analysis. The low number of glycolysis-related genes in these networks may indicate that glycolysis regulation is largely unchanged between the two conditions.

CD8^+^ Temra cells had the largest network of GTRD-annotated edges with significant difference in edge weights for all the comparisons.

The following centrality measures were calculated for each network: in-degree centrality, out-degree centrality, betweenness centrality, hub score and authority score. Since only transcription factors have both outgoing and incoming edges, centrality measures that depend on outgoing edges, such as out-degree centrality, hub score and betweenness centrality were considered only for the transcription factors. The centrality measures for the nodes in each network are provided in the Appendix A. We then identified the transcription factors that were present in the top ten out-degrees, hub scores and between centrality lists, as well as the target genes that were present in the top ten in-degree and authority score lists for each network. These are summarized in Table 15.

CD4^+^ Tem cells had several edges that showed a significant difference in edge weights in comparisons between both untreated RA vs. healthy samples and three treated vs. untreated RA samples. Among the transcription factors, *SPI1*, *PADI2*, *ZNF366* and *CBFA2T3* were in the top ten transcription factors for hub score, betweenness centrality and out-degrees for all the comparisons. Polymorphisms in the *PADI2* genes are associated with RA [33,34]. Its role in circulating CD4^+^ T cells of RA patients is unknown. *PPARG* and *VDR* were only present in the top TFs for IFX-treated samples, and *ETV1*, *NR2F2*, *NFE2* and *RBBP4* were present only for TCZ-treated samples. No target genes were common to the top ten target genes of all the comparisons. These differences may indicate changes in the regulation of glycolysis under different conditions in this cell type. Among the three treatments, TCZ showed the largest number of significant edges annotated with GTRD, with 4205 interactions between 1035 genes. This was also greater than the number of edges in the untreated RA vs. healthy samples network.

In CD8^+^ Tn cells, *FOXP1* was a top ten transcription factor across the transcription factor centralities in all the comparisons. FOXP1 is essential for maintaining quiescence in CD8^+^ Tn cells [35]. Its altered co-expression with glycolysis genes in RA CD8^+^ Tn cells may reflect changes in the metabolic repression associated with quiescence. *ALDH3B1*, *PCK1*, *ACSS2* and *ENO3* were the top ten target genes in authority score and in-degree for all the comparisons.

CD8^+^ Tcm cells had less than 20 target genes for all three treatment networks.

CD8^+^ Tem cells had *MBL2* as a top transcription factor and *GAPDH* as a top target gene in all the comparisons. *MBL2* polymorphisms are associated with RA in some ethnicities [36]. MTX had the largest number of significant GTRD-annotated edges among all three treatments, with 15,138 edges between 1248 genes. The size of the MTX vs. untreated RA network and the untreated RA vs. healthy network were very similar (Table 14), while the TCZ and IFX networks were smaller.

CD8^+^ Temra cells had the largest networks of differentially co-expressed genes among all the cell types. *ZNF143* was a top transcription factor in all the networks. *ENO1*, *G6PC3*, *ALDOC*, *ACSS2* and *HK2* were common top target genes. *TEAD1* was a top TF in all three treatments, but not in untreated RA.

### 3.7. GeneHancer Annotation of Transcription Factor–Target Gene Edges in Untreated RA CD8^+^ Tem Cells

The differentially co-expressed transcription factor–glycolysis gene edges in the untreated RA CD8^+^ Tem cells were annotated with enhancer binding site information from the database GeneHancer. This network had 281 nodes and 614 edges. Fifty-seven glycolysis-related genes were present in this network, eleven of which were not present in the GTRD-annotated network. Therefore, these genes had binding sites only in enhancer regions for the transcription factors in this study. The top ten glycolysis genes ranked by in-degree in the GeneHancer-annotated and GTRD-annotated differential co-expression networks for untreated RA CD8^+^ Tem cells are shown in Table 16.

The three differentially regulated genes—*PFKFB3*, *DLD* and *GAPDH*—show high centrality in both the networks. This indicates the potential for transcriptional regulation by a large number of transcription factors binding in both promoter and enhancer regions. The genes *FBP2*, *ALDOB*, *G6PC2* and *HKDC1* have a higher number of interactions in the GeneHancer network than in the GTRD network. However, the differential expression levels of these genes are low (Appendix A). *G6PD*, *SLC2A1* and *G6PD* are high-centrality genes in both networks, although they are not differentially expressed. *BPGM*, *PDHA1*, *PGAM1* and *AKR1A1* have high in-degrees in the GTRD network, but not in the GeneHancer network.

The three differentially regulated high-centrality genes *PFKFB3*, *DLD* and *GAPDH* shared a relatively high number of edges between the GTRD and GeneHancer networks. This means that the same set of transcription factors bind to sites in both promoter and enhancer regions of these genes. In order to compare the number of TF binding sites for specific TFs in the two databases, we plotted the number of binding sites for the common TF edges between the two. Figure 3 shows the bar plots of GTRD and GeneHancer binding sites for these TFs with *PFKFB3*, *DLD* and *GAPDH*.

In general, the number of binding sites in the GeneHancer were equal to or higher than the number of binding sites in GTRD. Table 17 shows the comparison of the numbers of binding sites in GTRD and GeneHancer. *PFKFB3* especially has several transcription factors with a high number of enhancer binding sites in GeneHancer (Figure 3). This may indicate a greater potential for transcriptional regulation via transcription factor binding to enhancer regions for *PFKFB3*.

### 3.8. Glycolysis Regulators in Untreated RA CD8^+^ Tem Cells

The CD8^+^ Tem cells from untreated RA individuals show increased expression of the glycolysis enzyme *GAPDH* and reduced expression of *DLD* when compared to healthy samples. Two key regulatory enzymes, *PFKFB3* and *PFKFB4*, show opposite differential expressions, with *PFKFB3* being downregulated and *PFKFB4* upregulated. *PFKFB3*, *GAPDH* and *DLD* are also genes with high target centrality scores, indicating that their correlation with transcription factors is significantly different in RA compared to healthy CD8^+^ Tem cells. Figure 4 shows the subgraph of *GAPDH*, *DLD* and *PFKFB3* with differentially expressed transcription factors in the GTRD-annotated network. Figure 5 shows the same in the GeneHancer-annotated network.

The differentially expressed transcription factors—*ZSCAN29*, *AEBP2*, *PHF20*, *IKZF5* and *ZNF24*—have edges with *GAPDH* in the GeneHancer network, which are not present in the GTRD network. *DLD* has an edge with *RUNX3* that is not present in the GTRD-annotated network. *PFKFB3* does not have any additional edges in the GeneHancer network compared to those present in the GTRD network.

The transcription factors that have significantly different edge weights with both *PFKFB3* and *PFKFB4* in the RA CD8^+^ Tem cells GTRD-annotated network and GeneHancer-annotated network are shown in Figure 6.

The transcription factors—*ATF3*, *E2F7* and *DTL*—have edges with both *PFKFB3* and *PFKFB4* in the GTRD-annotated network. *ATF3* and *E2F7* are also transcription factors with high centrality scores. They have edges with both *PFKFB3* and *PFKFB4* in the GeneHancer-annotated network, indicating the presence of binding sites in promoter and enhancer regions. *ATF3* and *E2F7* are known to regulate glucose metabolism [37,38,39] All transcription factors in both the networks have higher edge weights with *PFKFB4* in RA samples and higher edge weights with *PFKFB3* in healthy samples.

The differential regulation of other glycolysis regulators is shown in Table 18.

CD8^+^ Tem cells from tocilizumab-treated individuals showed an over-expression of *PCK2* and under-expression of *GAPDH*, two glycolysis-related genes. The gene *MYC*, an important regulator of glycolysis, is also upregulated in the samples, although it does not have high centrality scores. Figure 7 shows the subgraph of *GAPDH* and *PCK2* with differentially expressed transcription factors in TCZ-treated CD8^+^ Tem cells.

### 3.9. GeneHancer Annotation of Transcription Factor–Target Gene Edges in Untreated RA CD8^+^ Temra Cells

The GeneHancer-annotated, RA-untreated CD8^+^ Temra differential co-expression network has 500 nodes and 3478 edges. Seventy-three glycolysis genes were present in this network, three of which were not present in the GTRD-annotated network. Table 19 shows the top ten target genes ranked by in-degree in the GTRD-annotated and GeneHancer-annotated networks of untreated RA CD8^+^ Temra cells.

The differentially regulated genes, *ENO3* and *HK2*, show a high centrality in both the networks, while *PDHA1* has a high centrality in the GTRD network, but not in the GeneHancer network. The genes *G6PC2* and *ADH7* have a higher number of interactions in the GeneHancer network than in the GTRD network. However, the differential expression levels of these genes are low (Appendix A). *ENO1*, *G6PC3*, *AKR1A1*, *SLC2A1*, *HK1* and *ACSS2* are high-centrality genes in both networks, although they are not significantly differentially expressed. Some of these genes (*AKR1A1* and *HK1*) show clear changes in the expression levels, although they fall below the fold change cutoff (Table 20).

*AKR1A1* is not known to be associated with RA. However, it is known to be involved in nitric-oxide-based signalling, which is a major player in the lymphocytes of RA patients [40,41].

### 3.10. Glycolysis Regulators in CD8^+^ Temra Cells

CD8^+^ Temra cells of untreated RA samples showed the largest number of DEGs from the glycolysis pathway (Table 6). Among these genes, *ENO3* and *HK2* also show high target centrality scores. Among the transcription factors with high centrality scores, *MYC* was upregulated with a linear fold change of 3.14 and adjusted *p* value of 0.0004. Two other known regulators of glycolysis also show upregulation, but do not have high centrality scores:Insulin receptor (*INSR*): linear fold change of 3.77 and adjusted *p* value of 0.09;*PPARG*: linear fold change of 22.09 and adjusted *p* value of 0.01.

Figure 8 shows the interactions of *MYC*, *INSR* and *PPARG* with their target genes in the GTRD-annotated network.

The targets of *MYC* in this network include *ENO3*, *HK2*, *DLD* and *PDHA1*. *ENO3*, *HK2* and *PDHA1* have reduced edge weights with *MYC* in the RA samples compared to healthy CD8^+^ Temra cells, while *DLD* have higher edge weights in the RA samples.

*PPARG* and *MYC* have edges with *HK2*, *ENO3* and *PDHA1*, which are differentially regulated. *INSR* does not have edges with target genes that are differentially expressed; however, it has an edge with *MYC*. *MYC* also interacts with PPARG, and the differentially expressed glycolysis gene *DLD*.

Figure 9 shows the target glycolysis target genes of *MYC* and *PPARG* in the GeneHancer-annotated network for the untreated RA CD8^+^ Temra cells.

The targets of *MYC* and *PPARG* in the GTRD and GeneHancer networks are largely similar, indicating binding sites for these transcription factors in the cis regulatory regions outside the GTRD interval. On the other hand, *INSR* does not have any target genes in the GeneHancer network.

The over-expression of *MYC*, its high centrality scores and interactions with glycolysis-related genes and regulatory proteins that show differential expression in both the GTRD and the GeneHancer-annotated networks point to the central role of *MYC* in the altered glycolysis regulation of RA CD8^+^ Temra cells.

Among other genes involved in the regulation of glycolysis in lymphocytes, those showing a differential expression in this cell type are shown in Table 21.

### 3.11. Comparison of Untreated RA and TCZ-Treated CD8^+^ Tem Cells

CD8^+^ Tem cells from TCZ-treated individuals showed a downregulation of *GAPDH*, which was upregulated in untreated RA CD8^+^ Tem cells. Most other genes that were DEGs in untreated RA CD8^+^ Tem cells did not show significant differential expression in TCZ CD8^+^ Tem cells. Figure 10 shows the normalized mean counts of these genes in healthy, untreated RA and TCZ-treated CD8^+^ Tem cells.

*GAPDH*, *ATP5MG* and *COX8A* were significantly differentially regulated in both untreated RA and TCZ-treated CD8^+^ Tem cells. For these genes, the expression was increased relative to controls in the untreated RA samples, while their expression was reduced to levels close to that of controls in the TCZ-treated samples. The genes PCK2 and MYC were significantly upregulated in TCZ-treated samples relative to untreated RA. Their differential regulation in untreated RA relative to healthy samples was not significant. All remaining genes showed a significant differential regulation only in untreated RA CD8^+^ Tem cells relative to healthy cells. *ATP5F1C*, *ATP6V1G3*, *COX6A1*, *DLD*, *IDH3A*, *IDNK*, *EGLN3*, *ELOB*, *MT-CO1*, *NDUFA12*, *NDUFB10*, *NDUFB11*, *NDUFS2*, *OGDH*, *PFKFB4* and *UQCRHL* showed a non-significant return to healthy levels in the TCZ-treated CD8^+^ Tem cells. Four genes *TALDO1*, *PFKFB3*, *IRS2* and *RUNX3* show similar values in untreated RA and TCZ-treated CD8^+^ Tem cells. The expression levels of genes from glycolysis, the pentose phosphate pathway, TCA cycle and oxidative phosphorylation complexes 1, 4 and 5 in the healthy, untreated RA and TCZ-treated CD8^+^ Tem cells are shown in Appendix A.

### 3.12. Comparison of Untreated RA CD8^+^ Tem Cells and CD8^+^ Temra Cells

The cell types CD8^+^ Tem cells and CD8^+^ Temra show differential regulation of genes coding for metabolic enzymes and regulators of glycolysis. A different set of genes shows this alteration in each cell type. In Figure 11, the normalized counts of the genes that show differential regulation in either CD8^+^ Tem cells or CD8^+^ Temra cells of untreated RA are shown.

Table 6, Table 8, Table 9, Table 18 and Table 21 show the genes that are differentially regulated in CD8^+^ Tem and CD8^+^ Temra untreated RA cells relative to healthy cells. *DLD* and *EGLN3* are significantly differentially regulated in both cell types. Among the genes that are upregulated in either one of the cell types, *AKT1S1*, *ATP5F1C*, *ATP6V1G3*, *CAMKK2*, *COX6A1*, *COX8A*, *ELOB*, *ENO3*, *EP300*, *GAPDH*, *HIF1AN*, *HK2*, *IDH3A*, *IDNK*, *MT-CO1*, *NDUFA12*, *NDUFB10*, *NDUFB11*, *NDUFS2*, *NR1H2*, *OGDH*, *PDHA1*, *PFKFB3*, *PFKFB4*, *RUNX3* and *UQCRHL* all show similar trends in both cell types. *ATP5MG*, *HKDC1*, *INSR*, *IRS2*, *MYC*, *PPARG*, *TALDO1* and *RORC* show either opposing trends or do not show much difference in one of the cell types. The expression levels of genes from glycolysis, the pentose phosphate pathway, TCA cycle and oxidative phosphorylation complexes 1, 4 and 5 in the healthy, untreated RA and TCZ-treated CD8^+^ Tem cells are shown in the Appendix A.

### 3.13. RA CD8^+^ Tem Differentially Expressed Genes in PI3K-AKT Signalling Pathway and mTOR Signalling Pathway

The DEGs of RA CD8^+^ Tem cells were mapped to KEGG pathways using the KEGG color pathway tool. Figure 12 shows the DEGs in the PI3K Akt signalling pathway.

In the chemokine signalling part of this pathway, *GNG7* is downregulated. *COL6A5*, *ITGA1*, *ITGB6* and *PTK2* (FAK) are upregulated in the focal adhesion part of this pathway. *IFNA1*, *IFNA5*, *IFNA13* and *IL2RA* in the cytokine–cytokine receptor interaction part of the pathway are upregulated. In the growth factor–receptor tyrosine kinase part of the pathway, *EFNA1*, *EFNA4*, *VEGFC*, *ANGPT1*, *FLT1*, *KDR*, *RAS* are upregulated, while *FGF9* and *FLT4* are downregulated. Signalling via these pathways leads to the activation of PI3 kinase, which produces the mediator PIP3. Among the direct targets of AKT kinases, *eNOS* is upregulated and *CDKN1B* is downregulated. Other DEGs downstream of these direct targets include *CDK2* and *CCND3*, which are upregulated.

CD8^+^ T cells in RA patients are known to have activated mTOR [42]. The DEGs that are mapped to the mTOR signalling pathway are shown in Figure 13.

Wnt signalling leads to the inhibition of GSK3B and subsequent mTORC1 activation has four DEGs: *WNT11* is downregulated, while *WNT9A*, *WNT3A* and *FZD6* are upregulated. *HRAS* and *RPS6KA1*, which act downstream of the insulin signalling pathway, are upregulated. The mTOR activation in response to amino acids includes two upregulated genes: *ATP6V1G3* and *LAMTOR2*.

### 3.14. RA CD8^+^ Temra Differentially Expressed Genes in PI3K-AKT Signalling Pathway and mTOR Signalling Pathway

The DEGs of RA CD8^+^ Temra cells were mapped to KEGG pathways using the KEGG color pathway tool. Figure 14 shows the DEGs in the PI3K Akt signalling pathway.

Among the genes that lead to activation of the phosphatidylinositol-4,5-bisphosphate 3-kinase (PI K) enzymes, *IFNA21*, *IFNA10* and *IL6R* are activated in the cytokine–cytokine receptor interaction part of the pathway. This interaction leads to signalling via the JAK-STAT pathway resulting in the activation of PI3K. The receptor tyrosine kinases *ERBB3*, *FLT1* and *INSR* are upregulated. These activate PI3K via the IRS1 protein. The integrin genes *ITGA1* and *ITGA3* are upregulated. They activate the FAK kinase, which in turn, activates the PI3 kinase. In the chemokine signalling branch of the pathway, *LPAR3* and *GNG3* are upregulated. PI3 kinase generates PIP3 (phosphatidylinositol 3,4,5-trisphosphate), which activates the AKT kinases. *THEM4*, an inhibitor of AKT kinases, is upregulated in RA CD8^+^ Temra cells.

Among the direct targets of the AKT kinases, *eNOS* is upregulated, while *CREB3* is downregulated. The direct targets of the AKT kinases then activate or inhibit several downstream proteins. Among these, *MYC* and *CCND3* are upregulated, while *REL* and *RELA* are downregulated.

Figure 15 shows the RA CD8^+^ Temra DEGs in the mTOR signalling pathway.

This pathway shows upregulation in the Wnt signalling that leads to inhibition of the mTORC1 inhibitor complex of TSC1/2 and TBC1D7. The upregulated genes include *WNT7a* and *LRP5*. *RPS6KA1*, which performs the same function downstream of the insulin receptor *INSR*, is also upregulated. At the same time, the inhibitor of mTOR, *AKT1S1* is downregulated.

## 4. Discussion

In Table 2, we note that the CD8^+^ Tem and CD8^+^ Temra cells showed the greatest number of differentially expressed genes when comparing untreated RA and healthy samples.

Among the samples from treated RA individuals (Table 2 and Table 3), TCZ-treated samples showed a greater number of differentially expressed genes than MTX and IFX when compared with untreated RA individuals. CD8^+^ Tn and CD8^+^ Tem cells showed greater differential expression than CD8^+^ Temra cells in the TCZ-treated individuals. The reverse trend was experienced in the MTX- and IFX-treated individuals, where the CD8^+^ Temra cells showed the greatest number of differentially expressed genes among all the cell types.

### 4.1. Differential Expression of Metabolic Enzymes in CD8^+^ Tem Cells

Glycolysis/gluconeogenesis pathway enzymes were differentially expressed in CD8^+^ Tem and CD8^+^ Temra cells, when comparing untreated RA with healthy samples. The enzyme *GAPDH* was upregulated, while *DLD* was downregulated, in CD8^+^ Tem. *DLD* was also downregulated in CD8^+^ Temra cells. *ENO3*, *HK2*, *HKDC1* and *PDHA1* were upregulated in this cell type.

Figure 16 summarizes the differential expression of genes associated with metabolism in RA CD8^+^ Tem cells. Glycolysis, TCA cycle and oxidative phosphorylation have differentially expressed genes.

Based on the presence of genes and the functions of the genes in these pathways, we can conclude the following:The level of glycolysis in RA CD8^+^ Tem cells may not be different from that in healthy CD8^+^ Tem cells.The pentose phosphate pathway may be upregulated in the RA CD8^+^ Tem cells.There are both upregulated and downregulated genes in the TCA cycle, hence the effect of RA on CD8^+^ Tem cells is ambiguous.In oxidative phosphorylation, the genes of complex 1 are upregulated, while the catalytic component of the F0F1 ATP synthase is downregulated. Thus, there may be an increased activity of complex 1 and reduced ATP production.

Each pathway is discussed in more detail in the following sections.

### 4.2. CD8^+^ Tem Cells: Glycolysis

Glyceraldehyde-3-phosphate dehydrogenase (GAPDH) is a glycolysis pathway enzyme that converts glyceraldehyde-3-phosphate (GAP) to 1,3-bisphosphoglycerate utilizing NAD^+^ in the process. GAPDH is an important regulator of aerobic glycolysis in activated CD8^+^ Tem cells. On stimulation of the T cell receptor, the expression of cytoplasmic GAPDH increased, resulting in the switch to aerobic glycolysis [43]. GAPDH expression also increased in the hypoxic synovial lining of RA patients, and the response to TNF inhibitor treatment is accompanied by a reduction in GAPDH levels [31]. The inhibition of GAPDH by the immunomodulatory drug dimethyl fumarate has been shown to reduce the aerobic glycolysis of activated lymphoid cells [44].

*GAPDH* also has high centrality scores in the network of differentially co-expressed gene pairs, indicating a considerable change in its regulation in the RA CD8^+^ Tem cells compared to healthy cells.

However, it is not possible to infer an increase in aerobic glycolysis in RA CD8^+^ Tem cells compared to healthy cells purely based on the upregulation of *GAPDH*. The gene encoding the enzymes that catalyse the three rate limiting steps of glycolysis, namely, hexokinases (*HK 1-3*, *GCK*), phosphofructokinases (*PFKM*, *PFKL*, *PFKL*) and pyruvate kinases (*PKLR*, *PKM*) do not show any differential regulation in the RA CD8^+^ Tem cells. There is also no increase in the mRNA levels of glucose transporters which are required for the entry of glucose into the cells. Another indicator of aerobic glycolysis in T cells is the increased expression of lactate dehydrogenase A(LDHA). LDHA was previously found to be elevated at both mRNA and protein levels in all subsets of CD8^+^ T cells in RA patients compared to controls [8]. In this study, however, none of the T cell subsets showed differential expression of *LDHA*.

GAPDH becomes the rate limiting step in glycolysis only under the condition that its metabolites upstream in the glycolysis pathway are abundant and those downstream are scarce [45]. Since this study is only based on RNA seq data, we cannot comment on the levels of metabolites in the cells. GAPDH also has non-glycolytic functions, including the regulation of translation of genes, such as *HIF1A*, *MYC*, *IFNG*, *IL2* and *TNF*-α [46]. Thus, the increased expression of *GAPDH* does not necessarily predict an increase in aerobic glycolysis in RA CD8^+^ Tem cells; instead, any of its several other functions may be affected.

### 4.3. CD8^+^ Tem Cells: TCA Cycle

Three TCA cycle enzymes show differential regulation in CD8^+^ Tem cells, *DLD*, *IDH3A* and *OGDH*. The enzyme dihydrolipoamide dehydrogenase (*DLD*), which catalyses the conversion of dihydrolipoamide to lipoamide in an NAD^+^-dependent manner, is downregulated. DLD forms the E3 component of three enzyme complexes, namely, the pyruvate dehydrogenase complex (PDC), the α-ketoglutarate dehydrogenase complex (OGDC) and the branched-chain α-ketoacid dehydrogenase complex (BCKDC). In addition, DLD also takes part in the glycine cleavage system, which causes the catabolism of glycine [47]. All these enzyme complexes are located in the mitochondrion.

The genes *OGDH* and *PDHA1* transcribe enzymes that form part of the large complexes OGDC and PDC. The RA CD8^+^ Tem cells show an upregulation of *OGDH*. While *PDHA1* upregulation is significant (FDR *p* value 0.01980018), the linear fold increase is only 1.43.

The OGDC catalyses the irreversible decarboxylation of α-ketoglutarate to succinyl CoA, generating NADH in the process. This reaction is a major point of regulation of the citric acid cycle and is tightly controlled by various means [48]. The OGDC consists of several copies of three enzymes: α-ketoacid decarboxylase (E1), dihydrolipoyl transacetylase (E2) and dihydrolipoamide dehydrogenase (E3). The upregulated OGDH gene codes for the E1 component, while DLD codes for the E3 subunit [49]. The product of OGDC, succinyl-CoA, is converted to succinate by the enzyme succinyl-CoA synthetase. This enzyme consists of two subunits, α encoded by SUCLG1 and β, encoded by SUCLG2 or SUCLA2. Activated CD4^+^ Tn cells from the peripheral blood of RA patients show reduced mRNA levels of *SUCLG2* and *DLD*, as well as reduced protein levels and enzymatic activity of succinyl-CoA synthetase [32]. This leads to a reversal of the citric acid cycle, causing the accumulation of acetyl CoA, citrate and α-ketoglutarate. This is known as a “broken” TCA cycle.

The expression levels of *SUCLG1*, *SUCLG2* and *SUCLA2* are not altered in any cell type in this study. DLD expression is also seen to be reduced only in CD8^+^ Tem and CD8^+^ Temra cells.

*IDH3A*, which encodes the catalytic subunit of the NAD^+^-dependent isocitrate dehydrogenase enzyme, is mildly upregulated. This enzyme catalyses the conversion of isocitrate to α-ketoglutarate in the citric acid cycle [50]. This step precedes the OGDC catalysed reaction in the TCA cycle. IDH3 is found to be active in the RA CD4^+^T cells with a “broken” TCA cycle [32]. Therefore, the increased expression of *OGDH* and *IDH3A* by themselves do not necessarily indicate an intact TCA cycle in CD8^+^ Tem cells. However, the lack of downregulation of any TCA cycle enzyme except *DLD* may mean that any TCA change in the regulation of the TCA cycle in the CD8^+^ RA Tem cells does not occur at the transcription level.

The PDC is composed of three enzymes: pyruvate dehydrogenase (E1), dihydrolipoamide acetyltransferase (E2), and dihydrolipoamide dehydrogenase (E3) and a fourth component E3BP. The *PDHA1* gene codes for the E1 component of the complex [51]. The PDC converts pyruvate into acetyl CoA, generating NADH from NAD^+^ in the process. Acetyl-CoA is then utilized in the citric acid cycle, and the NADH enters oxidative phosphorylation [52]. The PDC thus links glycolysis with the citric acid cycle and oxidative phosphorylation.

Aerobic glycolysis involves conversion of pyruvate to lactate rather than acetyl CoA. This leads to reduced availability of pyruvate for the citric acid cycle. In these conditions, pyruvate can be generated by glutamine to replenish the TCA cycle in T lymphocytes [53]. Thus, the presence of aerobic glycolysis does not preclude a functional TCA cycle.

It is not possible to comment on the increase or decrease in the activity of the PDC in RA CD8^+^ Tem cells based on the expression levels of PDHA1 and DLD, since they show discordant changes. Among the post translational regulators of the PDC, neither the pyruvate dehydrogenase kinases (*PDK1-4*), which inhibit PDC activity, nor the pyruvate dehydrogenase phosphates (*PDP1/2*, *PDPR*), which reactivate the PDC, show any differential regulation [52,54]. Therefore, although some TCA cycle enzymes show differential expression, it is not possible to conclusively state that the TCA cycle in RA CD8^+^ T cells is increased or decreased.

### 4.4. CD8^+^ Tem Cells: Oxidative Phosphorylation

Eleven genes coding for proteins in the oxidative phosphorylation pathway are differentially regulated, with all but one (*ATP5F1C*) being upregulated. *ATP5F1C* codes for the γ subunit of the catalytic F1 head of the mitochondrial F0F1 ATP synthase [55]. The downregulation of this gene is a clear indication of reduced ATP production by oxidative phosphorylation in RA CD8^+^Tem cells compared to healthy CD8^+^Tem cells.

However, genes coding for proteins in complex I (*NDUSF2*, *NDUFA12*, *NDUFB10*, *NDUFB11*) are upregulated. NDUFS2 is the core subunit of complex 1 and responsible for electron transfer from NADH to ubiquinone [56]. The counts of the other differentially expressed genes in this pathway are low and may be biologically insignificant. Thus, RA CD8^+^ Tem cells may have greater complex 1 activity than their healthy counterparts. Complex 1 activity is essential for the function of CD8^+^T cells [57].

CD8^+^ Tem cells from RA patients thus show increased expression of three TCA cycle enzymes and complex 1 enzymes. Among these, OGDC and complex 1 also generate reactive oxygen species [58,59]. The over-expression of genes coding for these enzymes may lead to increased ROS production in RA CD8^+^ T cells compared with healthy controls. This is in contrast to CD4^+^ T cells in RA, which are known to have a pathological reductive phenotype [60].

### 4.5. CD8^+^ Tem Cells: Regulation of Glycolysis and the Pentose Phosphate Pathway

Figure 17 describes the possible changes to the regulation of glycolysis in RA CD8^+^ Tem cells.

CD8^+^ Tem cells show decreased expression of *PFKFB3* and increased expression of *PFKFB4*, two isozymes of the bifunctional enzyme that control the concentration of fructose 2,6-bisphosphate. Fructose 2,6-bisphosphate is an allosteric modulator of phosphofructokinase (PFK) that catalyses the commitment step of glycolysis. Although both PFKFB3 and PFKFB4 can catalyse the production and destruction of fructose 2,6–bisphosphate, PFKFB3 has greater kinase activity and favours glycolysis. PFKFB4, on the other hand, favours flux through the pentose phosphate pathway [61]. It has been shown that, for CD4^+^T cells in RA, *PFKFB3* is under-expressed, causing the increased activity of the pentose phosphate pathway, which in turn leads to greater NADPH production. The NADPH is utilized for the generation of fatty acids and sets up a reductive environment in the cell [7]. CD8^+^ T cells also express less PFKFB3 than controls at the protein level [8]. Two pentose phosphate pathway enzymes are upregulated, *TALDO1* and *IDNK*. The shift to the pentose phosphate pathway depends on the ratio of PFKFB3 to the enzyme G6PD, which catalyses the first step of the pentose phosphate pathway [7]. Thus, a reduction in PFKFB3 levels may be sufficient to drive this shift even in the absence of an increase in the expression of G6PD.

Among the known regulators of aerobic glycolysis in CD8^+^T cells, *HIFA* expression does not significantly change. However, two of the negative regulators of HIF1A in normoxia, *EGLN3* and *ELOB*, are upregulated [62,63]. Since circulating CD8^+^ T cells in RA do not experience hypoxia, it can be assumed that HIF1A remains inhibited in these cells. In addition, GAPDH also binds to and represses the translation of the HIF1A mRNA. HIF1A induces the expression of several genes associated with glycolysis, including *PFKFB3* [64]. The downregulation of *PFKFB3* may be attributed to the inhibition of *HIF1A* in RA CD8^+^ Tem cells. Since the inhibition of *HIF1A* occurs post transcriptionally, the mRNA levels of *HIF1A* may remain unchanged, hence the edge weight between *HIF1A* and *PFKFB3* need not be significantly altered between RA and healthy CD8^+^ Tem cells.

*PFKFB4* expression is also induced by HIF1A [65]. The increased expression of *PFKFB4* may lead to greater pentose phosphate pathway activity, but the cause of this increased expression cannot be explained by the proposed inhibition of HIF1A.

Based on the differential expression of these genes and the altered edge weights of the TF-glycolysis gene pairs, we suggest that HIF1 α does not mediate aerobic glycolysis in RA CD8^+^ Tem cells.

*PFKFB3*, *GAPDH* and *DLD* also have high centrality scores (Table 15). The altered regulation of these genes may represent the change in regulation of the glycolysis pathway. *PATZ1* and *RUNX3*, the two transcription factors required for the differentiation and lineage integrity of CD8^+^T cells, are downregulated in the RA CD8^+^Tem cells [66]. Both genes also have reduced edge weights with *PFKFB3* in RA. At the same time *LEF1*, which also plays a role in maintaining CD8^+^ T cell identity, is upregulated, and its association with *PFKFB3* is increased in RA [67]. The change in the edge weights of *PFKFB3* with these genes may indicate a possible link between T cell lineage maintenance and the regulation of glycolysis in these cells. *PFKFB3*, *GAPDH* and *DLD* also have a high degree centrality in the GeneHancer-annotated network, indicating the presence of enhancer binding sites for several transcription factors (Table 16). *PFKFB3* also has a large number of binding sites for specific transcription factors with which it shows differential co-expression (Figure 3). Thus, transcription factors other than HIF1A may be involved in regulating *PFKFB3* expression in RA CD8^+^ Tem cells. PFKFB3 is also known to increase cell proliferation by inhibiting CDKN1B [68]. However, both *PFKFB3* and *CDKN1B* are downregulated in these cells. Thus, this potential mechanism of linking glycolysis with cell proliferation does not exist in the RA CD8^+^ Tem cells.

### 4.6. CD8^+^ Tem Cells: Transcriptional Regulation of GAPDH

Figure 18 shows the differentially expressed transcription factors which have edges with *GAPDH* in the RA CD8^+^ Tem cells. The functions of each transcription factor are also described.

Among the TFs that have edges with *GAPDH*, *MXI1*, *CDKN1B* and *KDM3B* are downregulated. MXI1 is an antagonist of MYC [69], and MYC is known to induce the transcription of glycolysis-related genes during the activation of T cells [70]. The downregulation of *MXI1* may relieve the inhibition of MYC function in RA CD8^+^ Tem cells. CDKN1B inhibits cell proliferation in CD8^+^T lymphocytes [71]. KDM3B may play a tumour-suppressor role in acute myeloid leukemia [72]. Their reduced expression indicates greater proliferative abilities for the RA CD8^+^ Tem cells. *KDM3B*, *MXI1* and *CDKN1B* have reduced edge weights with *GAPDH* in the RA CD8^+^ Tem cells, indicating a possible coordination of *GAPDH* levels with these processes.

The upregulated gene *PTBP1* is known to drive aerobic glycolysis by promoting the expression of the PKM2 splice isoform of the pyruvate kinase enzyme [73]. However, its association with *GAPDH* is not understood. CTBP1, a transcriptional repressor that antagonizes tumour suppressors, is activated by NADH and in turn increases the levels of NADH in cells. This positive feedback loop involves the repression of the mitochondrial pyruvate carriers MPC1 and MPC2 [74]. The MPCs transport pyruvate from the cytosol into the mitochondria where it becomes the substrate of PDC, and the inhibition of this process is associated with increased lactate production and aerobic glycolysis. *CTBP1* is upregulated in the RA CD8^+^ Tem cells; however, the targets of its repression, *MPC1* and *MPC2*, do not show downregulation. Thus, it cannot be concluded that pyruvate transport to mitochondria by the MPC1/2 proteins is reduced. The GAPDH-catalysed reaction also generates NADH. The increased edge weight between *GAPDH* and *CTBP1* in RA may indicate another mechanism by which *CTBP1* increases NADH levels in the cells.

*NOTCH1*, which is a promoter of aerobic glycolysis, was upregulated in CD8^+^ Tem cells [75]. However, its target genes in glycolysis were not differentially regulated. *NOTCH1* mRNA translation is repressed by GAPDH when GAPDH is modified by the metabolite methylglyoxal [76]. Methylglyoxal is generated from glyceraldehyde-3-phosphate, which is the substrate for GAPDH. This function of GAPDH occurs only when it is not taking part in glycolysis [43]. It is not possible to ascertain whether this inhibition of NOTCH1 translation by GAPDH occurs from transcriptome data alone.

*TP63*, which is upregulated, also has an increased edge weight with *GAPDH* in RA CD8^+^ Tem cells. ΔNp63α, a splice isoform of TP63, induces the expression of GAPDH in squamous cell carcinoma when phosphorylated [77]. On the other hand, another isoform, TAp63, may induce the pentose phosphate pathway as well as glycolysis [78,79]. Thus, the increased expression of the *TP63* gene may lead to the over-expression of *GAPDH*, depending on the splice isoform that is dominant in RA CD8^+^ Tem cells. In summary, GAPDH may play a central role in RA CD8^+^Tem cell function, but its role in the regulation of glycolysis is unclear.

### 4.7. Differential Regulation in the PI3K-AKT and mTOR Signalling Pathways in CD8^+^ Tem Cells

Figure 12 and Figure 13 show the differentially regulated genes in the PI3K-AKT signalling and mTOR signalling pathway. However, due to the presence of both up and downregulated genes in these pathways, the KEGG pathway diagram does not show a clear picture of the expression states of the gene in the pathway. Figure 19 shows all the DEGs in these pathways.

As seen in Figure 19, some important proteins such as *IRS2* are downregulated in the pathways. IRS2 activates PI3K activity in response to signalling from growth factor receptors, resulting in increased aerobic glycolysis [80]. This signalling cascade involves the activation of mTOR and cannot be fully compensated by IRS1 [80]. The differential regulation of *WNT11*, *WNT9A*, *WNT3A*, *FZD6* and *ATP6V1G3*, though significant, may not have any biological importance due to their low counts in the RA CD8^+^ Tem samples. Therefore, we cannot say with certainty that mtorc1 is activated based on the differential regulation of these genes.

The possible regulation of glycolysis metabolism in RA CD8^+^ Tem cells is summarised in Figure 20.

### 4.8. Effect of Treatment on the Regulation of Glycolysis in CD8^+^ Tem Cells

Tocilizumab is an antagonist of the IL 6 receptor used to treat rheumatoid arthritis [81]. This drug is shown to reduce the ability of RA T cells to proliferate [9]. In tocilizumab-treated CD8^+^ Tem cells, *GAPDH* is downregulated and *PCK2* is upregulated relative to untreated samples (Table 7). They also have high target centrality scores in the GTRD-annotated differential co-expression network of TCZ-treated RA CD8^+^ Tem cells (Table 15). PCK2 is a mitochondrial enzyme that catalyses the conversion of oxaloacetate to phosphoenolpyruvate, utilizing GTP. This reaction is the first step of the gluconeogenesis pathway [82]. *PCK2* is not differentially expressed in untreated RA CD8^+^ Tem cells, although there is a non-significant decrease in the mean counts relative to healthy samples. As PCK2 can control the flux through the TCA cycle, its upregulation may indicate that changes in RA CD8^+^ Tem cell TCA cycle by TCZ treatment may involve a return to a healthier state.

GAPDH downregulation in response to TCZ therapy may have several consequences, due to the varied functions of the gene. The upregulated transcription factors *STAT2*, *CRY1* and *ZEB1* have increased edge weights with *GAPDH* in the TCZ-treated CD8^+^Tem cells. STAT2 causes transcriptional activation in response to interferon signalling [83]. It is also known to induce the secretion of IL 6 [84]. *CRY1* is a circadian clock-related gene that also plays a part in arthritis. Mice lacking *CRY1* and *CRY2* have increased levels of activated T cells, and when arthritis was induced in these mice, serum IL6 levels were higher than in wild type mice with induced arthritis [85]. ZEB1 is involved in T cell development and is known to upregulate IL6 gene expression [86,87]. The relation between these genes and GAPDH is not clear.

Among the transcriptional regulators of glycolysis, *MYC* is upregulated, while *HIF1A* expression is unchanged. *ELOB*, an inhibitor of HIF1A that is upregulated in RA CD8^+^ Tem cells, is downregulated in TCZ-treated CD8^+^ Tem cells. *EGLN3* expression is also reduced, though it is not statistically significant (Table 18 and Figure 10). These changes, along with the decreased expression of *GAPDH*, remove the inhibition of HIF1A proposed in Figure 16. However, the *HIF1A* and *MYC* target genes that are involved in glycolysis do not show any differential expressions. In addition to this, the removal of HIF1A inhibition also does not result in an increase in the expression of *PFKFB3* caused by TCZ treatment (Figure 10). *PFKFB3* does not have any differentially co-expressed edges in the TCZ CD8^+^ Tem network. This indicates that there is no change in the mRNA levels of any transcriptional regulators of *PFKFB3* due to TCZ treatment. At the same time, the downregulation of the inhibitors of HIF1A function may point to the presence of other mechanisms for the continued downregulation of *PFKFB3* in the TCZ-treated CD8^+^ Tem cells.

As shown in Figure 10, most genes that are differentially regulated in untreated RA CD8^+^ Tem cells show a return to levels close to healthy mean counts in TCZ-treated samples. The exceptions include *TALDO1*, *PFKFB3*, *IRS2* and *RUNX3*. *PFKFB3* and *IRS2* are important regulatory genes for glycolysis. Thus, TCZ treatment does not bring the regulation of glycolysis in CD8^+^ Tem cells to a completely healthy state. Since *PFKFB3* levels were not changed, and do not have high centrality scores, the balance between glycolysis and the pentose phosphate pathway may not have altered relative to untreated RA CD8^+^ Tem cells. On the other hand, oxidative phosphorylation may be more similar to a healthy state in the TCZ-treated CD8^+^ Tem cells as shown by the genes *ATP5F1C*, *ATP6V1G3*, *COX6A1*, *MT-CO1*, *NDUFA12*, *NDUFB10*, *NDUFB11*, *NDUFS2*, *ATP5MG*, *COX8A* and *UQCRHL*.

### 4.9. CD8^+^ Temra Cells: Glycolysis

CD8^+^ Temra cells, also known as CD8^+^ Temra cells, are increased in the blood of RA affected individuals whose samples were used in this study [9]. RA CD8^+^ Temra cells show the over-expression of four glycolysis genes—*PDHA1*, *ENO3*, *HKDC1* and *HK2*. *HKDC1* and *HK2* counts in untreated RA and healthy cells are low, hence their over-expression may not have any biological effect. Similar to CD8^+^ Tem cells, RA CD8^+^ Temra cells also have high *PDHA1* and low *DLD*.

ENO3 (or β enolase) is a subunit of enolase that converts 2-phosphoglycerate to phosphoenolpyruvate in a reversible reaction [88]. β enolase forms homodimers or heterodimers with α or γ enolase to create a functional enzyme, and is predominantly found in the muscle [89]. Its role in RA is unknown.

### 4.10. CD8^+^Temra Cells: Regulation of Glycolysis

There is no clear indication of an increase in the expression of key glycolysis enzymes, except for *PDHA1* and *ENO3*. However, there is a large network of differentially co-expressed gene pairs, indicating changes in the regulation of glycolysis. Among the target genes with a high centrality, *G6PC3*, *SLC2A1* and *HK2* are important for regulating the glycolysis pathway.

*G6PC3* codes for a catalytic component of the glucose 6 phosphatase system, which catalyses the conversion of glucose 6 phosphate to glucose. This is the final step in glycogenolysis and gluconeogenesis. The deficiency of G6PC3 is known to cause neutropenia; however, its role in lymphocytes is unclear [90]. *SLC2A1* codes for the glucose transporter GLUT1, which is one of the proteins responsible for the increased aerobic glycolysis in several cancers [91]. However, the RA CD8^+^ Temra cells do not show an increase in the mRNA levels of *SLC2A1*. Another protein that shows increased expression during aerobic glycolysis is hexokinase2 (HK2) [92]. As discussed earlier, *HK2* upregulation may not be biologically significant in RA CD8^+^ Temra cells. However, *HK2* also has high target centrality scores, indicating changes in its transcriptional regulation. Thus, in spite of the changes in its transcriptional regulation, its upregulation may not have any relevance.

The transcription factors with high centrality scores include *MYC*, a known regulator of aerobic glycolysis in T cells [93]. MYC is known to increase the expression of HK2 and PDK1 [94]. PDK1 is an inhibitor of pyruvate dehydrogenase, which is also inducible by HIF1A during the process of aerobic glycolysis [95]. Both *HK2* and *PDK1* have reduced edge weights with MYC in the RA samples. The *HK2* levels were low in RA CD8^+^Temra cells despite a statistically significant increase compared to the healthy samples, and *PDK1* was not upregulated. *PDHA1* and *ENO3*, which were upregulated, have lower edge weights with *MYC* in the RA samples. These changes suggest that the upregulation of *PDHA1* and *ENO3* was not due to MYC binding in the RA CD8^+^Temra cells. In addition, the upregulation of *MYC* did not cause an increase in the expression of glycolysis genes.

Among other known regulators of glycolysis, *PPARG* and *INSR* are upregulated. Neither have high centrality scores. PPARG upregulates HK2 in medulloblastoma, and an agonist of PPAR signalling increases the rate of glycolysis in CD8^+^ cells [96,97]. In RA CD8^+^Temra cells, the edge weights between *PPARG* and *HK2*, *HK1*, *PFKP*, *LDHA*, *PDHA1* and *ENO1* are reduced, whereas the edge weight between *SLC2A1* and *ENO3* are increased in the RA samples. These changes indicate that *PPARG* upregulation is unlikely to drive any increase in glycolysis in RA CD8^+^ Temra cells.

The insulin receptor *INSR* is also upregulated; however, its edge weight with MYC is greater in healthy CD8^+^ Temra cells.

*HIF1A* is not differentially regulated in RA CD8^+^Temra cells, and it does not have high centrality scores. It has edges with only two glycolysis genes, *ALDOC* and *ADH7*. In both cases, the edge weights are higher in healthy CD8^+^ Temra cells. These results indicate that HIF1A may not regulate increases in glycolysis in RA CD8^+^ Temra cells compared to healthy cells. In addition, two of the suppressors of *HIF1A*, *EGLN3* and *HIF1AN* are upregulated in RA CD8^+^ Temra cells. The co-activator of HIF1A, *EP300* is also downregulated [98]. Thus, as in the case of CD8^+^ Tem cells, RA CD8^+^ Temra cells also may have a function of suppressing HIF1A. However, this suppression of HIF1A activity only leads to a small, non-significant downregulation of *PFKFB3* in RA CD8^+^ Temra cells (Figure 11).

Among the signalling pathways that control metabolism in T cells, PI3K-AKT signalling and mTOR signalling show differential regulation in CD8^+^ Temra cells. The signalling pathways upstream of the PI3 kinases show the upregulation of several genes, indicating a possible activation of the kinase and the generation of the second messenger PI3P (Figure 13). Only one of the inhibitors of the AKT kinases shows upregulation. Thus, the RA CD8^+^ Temra cells may have active signalling via the PI3K-AKT pathway. The downstream targets of AKT include TSC2, an inhibitor of the mTORC1 complex [26]. The mTORC1 complex is part of the mTOR signalling pathway, which also shows DEGs in the RA CD8^+^ Temra cells (Figure 15).

AKT phosphorylation activates important glycolysis-related genes, such as hexokinase, PFKFB2 and PDK1. In addition, it increases the plasma membrane localization of the GLUT1 and GLUT4 glucose transporters. In summary, AKT kinases increase the activity of the glycolysis pathway [26]. AKT signalling also leads to an increased transcription of HIF1A and MYC. However, HIF1A levels are unchanged in the RA CD8^+^ Temra cells. The glycolytic genes that are targets of MYC show upregulation; however, as discussed earlier, their edge weights with MYC are higher in the healthy controls, indicating that their upregulation may not be related to MYC levels.

The mTOR pathway shown in Figure 15 shows the upregulation of genes that cause the activation of the pathway and the downregulation of an inhibitor of the pathway. In addition to the genes shown in Figure 15, the PI3K-AKT pathway also leads to the activation of the mtorc1 complex. Figure 21 shows the relation between the two pathways, and the differential regulation in RA CD8^+^ Temra cells that may cause increased mtorc1 activity. The effect of mTOR signalling on glycolysis is mediated by *HIF1A* and *MYC* upregulation [27]. The activation of mTOR was demonstrated in the CD8^+^ T cells of RA patients [42].

In general, most of the considered genes that show significant differential expressions in RA CD8^+^ Tem cells show similar but non-significant differential expressions in the same direction in the RA CD8^+^ Temra cells (Figure 11). The exceptions to this trend include *MYC*, *INSR* and *IRS2*. Since these proteins are important regulators of glycolysis, the difference in their expression profiles may have profound effects on the regulation of glycolysis for both cell types. As discussed earlier, *MYC* does not have increased edge weights with its glycolysis targets genes in the RA samples, nor do any of these genes show large changes to their expression. However, the insulin receptor INSR and its substrate IRS2 exert their effects through post translational modifications of downstream proteins. Therefore, the effects of their altered differential regulation in the two cell types are not visible in the transcriptomic data. The changes in insulin signalling, as well as the PI3K-AKT and mTOR signalling pathways, indicate immediate changes in the glycolytic state of the RA CD8^+^ Tem and RA CD8^+^ Temra cells.

Figure 22 summarises the changes in glycolysis regulation in RA CD8^+^ Temra cells.

### 4.11. Limitations of This Study

This analysis is only based on gene expression data from the peripheral blood T lymphocyte subsets of healthy, untreated and treated RA individuals. Thus, information about activity levels of the glycolysis enzymes and concentration of metabolites cannot be included in this study. The signalling pathways discussed here use post transcriptional modifications and changes in localization to regulate glycolytic activity. This study cannot capture those changes. Transcription factors act as part of protein–protein interaction networks with other proteins, such as co-factors and chromatin remodellers, in order to initiate transcription [99]. Such interactions are also not part of the current analysis. However, this study is the most comprehensive analysis of the effect of RA on the transcriptional regulation of glycolysis in the T lymphocyte subsets present in circulation.

## 5. Conclusions

The differential expression of glycolysis-related genes and the differential co-expression of glycolysis-related genes with transcription factors were examined in healthy, untreated RA and treated RA T cell subsets. CD8^+^Tem and CD8^+^Temra cells showed the greatest number of differentially expressed genes and the largest networks of differentially co-expressed interactions, indicating changes in the transcriptional regulation of glycolysis. CD8^+^ Tem cells of untreated RA individuals a showed differential regulation and high centrality scores for *GAPDH* and *PFKFB3*. A model for the differential expression of *PFKFB3* by post transcriptional inhibition of HIF1A has been proposed. *GAPDH* may play a role in several non-glycolytic processes in RA Tem cells based on its association with differentially regulated transcription factors. TCZ-treated CD8^+^ Tem cells showed a partial reversal of the differential expression of oxidative phosphorylation, TCA cycle, pentose phosphate pathway and glycolysis genes, as well as some of the glycolysis regulators. CD8^+^ Temra cells of untreated RA individuals showed upregulation of PI3K-AKT and mTOR pathways, possibly leading to *MYC* over-expression. High centrality scores of *MYC* indicated changes in its mediated transcriptional regulation.

The effect of these changes on the activity of glycolysis genes must be examined experimentally. The role of GAPDH in RA CD8^+^ Tem cells can also be explored using protein–protein interaction networks.

## Figures and Tables

**Figure 1 genes-13-01216-f001:**
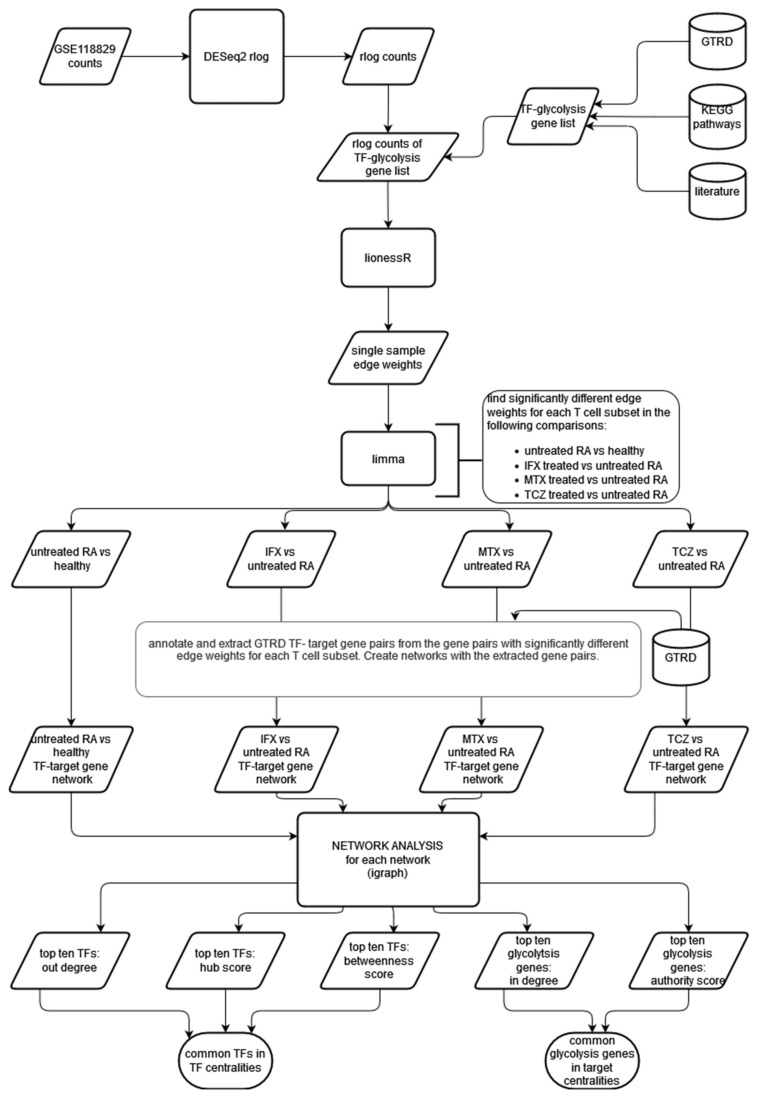
Overview of the workflow. TF—transcription factor; IFX—infliximab–methotrexate combination therapy; MTX—methotrexate monotherapy; TCZ—tocilizumab monotherapy.

**Figure 2 genes-13-01216-f002:**
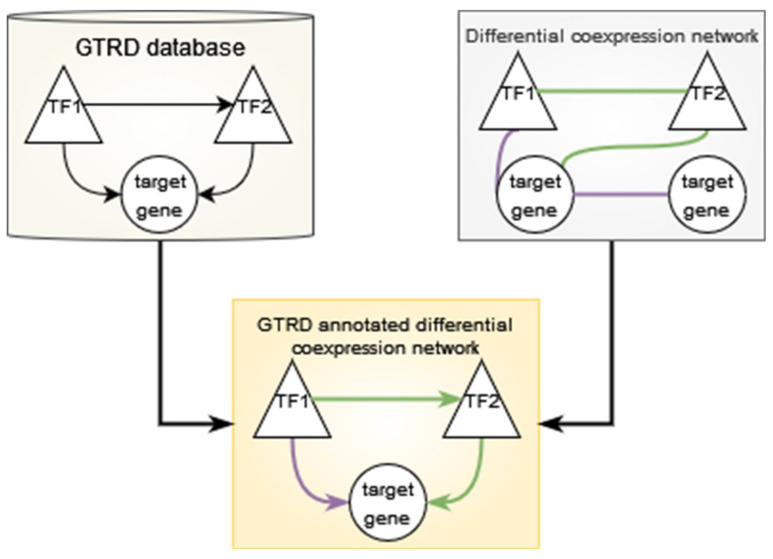
Construction of the GTRD-annotated differential co-expression network. The GTRD database contains transcription factors (triangles) and target genes (circles). The presence of a binding site for the transcription factor in the promoter region of the target gene is shown by an arrow pointing from the transcription factor (TF) to the target gene. A TF–TF interaction is also shown. The differential co-expression network has gene pairs that show significantly increased edge weights (purple connector) or significantly decreased edge weights (green connector) in the comparison. The connections are not directed (connector without arrow) and may occur between any two genes. The GTRD-annotated differential co-expression network has TF–TF edges or TF–target gene edges if the edge is present in both GTRD and the differential co-expression network. The direction for the edge is annotated using GTRD. Purple arrows show a TF–target gene or TF–TF interaction where the edge weight is increased, and green arrows show one where the edge weight is decreased. There are no target gene–target gene edges in this network.

**Figure 3 genes-13-01216-f003:**
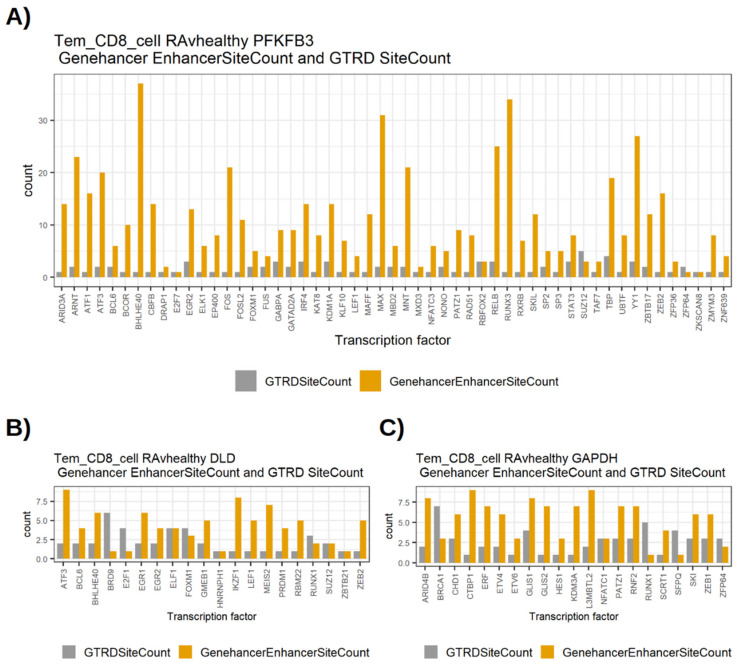
Number of transcription factor binding sites for transcription factors that interact with *PFKFB3*, *DLD* or *GAPDH*, as annotated by GTRD and GeneHancer. Grey bars represent the number of TF binding sites in the interval −1000 to +100 around the transcription start site of the target gene according to GTRD. Yellow bars represent the number of TF binding sites outside the interval −1000 to +100 around the transcription start site of the target gene according to GeneHancer. (**A**) The transcription factors binding sites for *PFKFB3*, (**B**) The binding sites for *DLD*, (**C**) The binding sites for *GAPDH*.

**Figure 4 genes-13-01216-f004:**
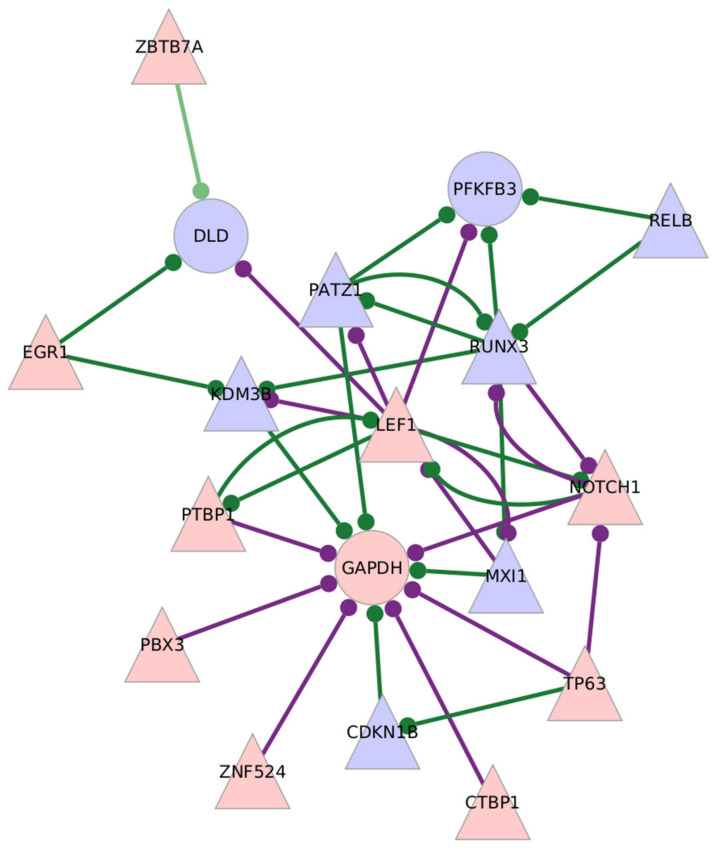
Subgraph of differentially expressed glycolysis-related enzymes with high centrality scores in the GTRD-annotated network of edges with significantly different edge weights between untreated RA and healthy control samples, in CD8^+^ Tem cells. The differentially expressed transcription factors that link to the enzymes are shown. Triangles indicate transcription factors, and circles indicate target genes. Node colour shows the direction of differential expression: red nodes are over-expressed and blue are under-expressed. The presence of an edge between a TF and a target gene refers to a significant difference in the edge weights of that gene pair between untreated RA samples and healthy samples. Purple edges show higher edge weight in RA, and green shows higher edge weight in healthy samples.

**Figure 5 genes-13-01216-f005:**
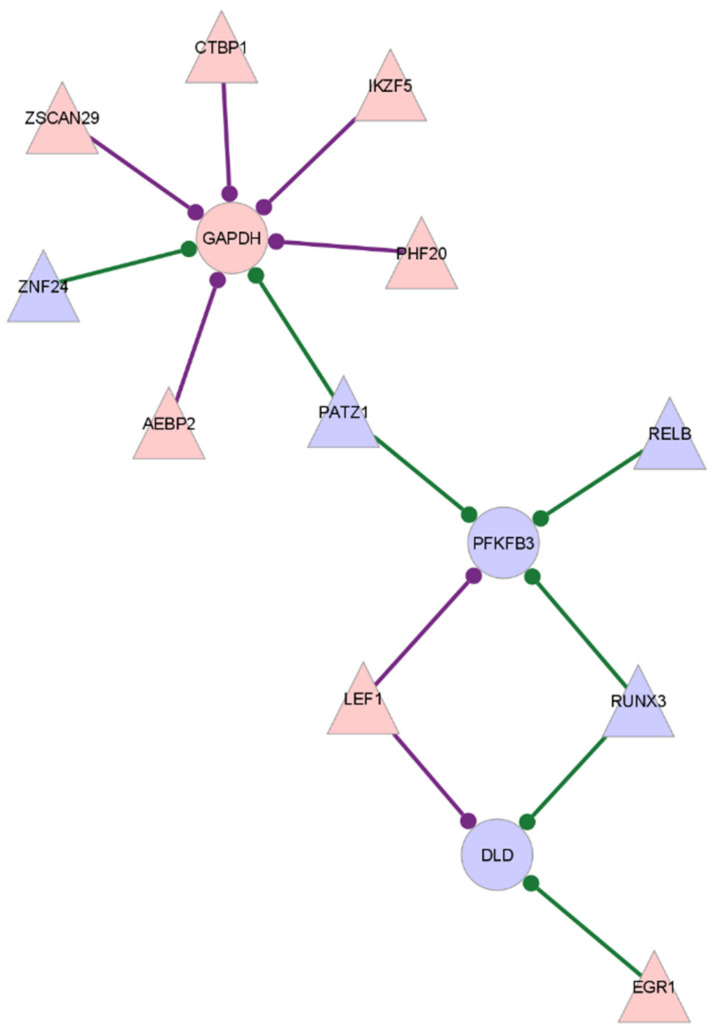
Subgraph of differentially expressed glycolysis-related enzymes with high degree centrality in the GeneHancer-annotated network of edges with significantly different edge weights between untreated RA and healthy control samples, in CD8^+^ Tem cells. The differentially expressed transcription factors that link to the enzymes are shown. Triangles indicate transcription factors and circles indicate target genes. Node colour shows the direction of differential expression: red nodes are over-expressed and blue nodes are under-expressed. The presence of an edge between a TF and a target gene refers to a significant difference in the edge weights of that gene pair between untreated RA samples and healthy samples. Purple edges show higher edge weight in RA, and green shows higher edge weight in healthy samples.

**Figure 6 genes-13-01216-f006:**
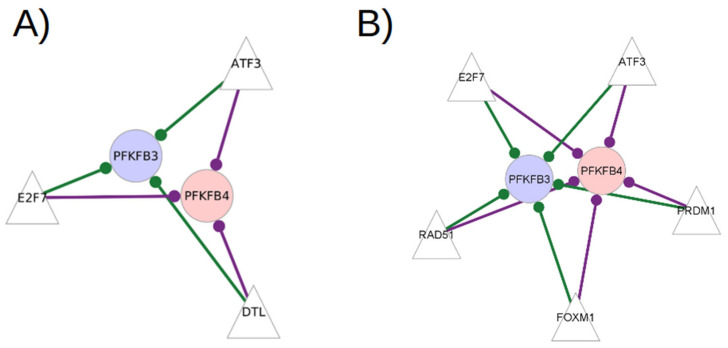
The transcription factors that have edges with both *PFKFB3* and *PFKFB4* in the RA CD8^+^ Tem cell GTRD and GeneHancer-annotated networks of differentially co-expressed edges. (**A**) The subgraph from the GTRD-annotated network (**B**) The subgraph from the GeneHancer-annotated network. Triangles indicate transcription factors and circles indicate target genes. Node colour shows the direction of differential expression: red nodes are over-expressed and blue nodes are under-expressed. The presence of an edge between a TF and a target gene refers to a significant difference in the edge weights of that gene pair between untreated RA samples and healthy samples. Purple edges show higher edge weight in RA, and green shows higher edge weight in healthy samples.

**Figure 7 genes-13-01216-f007:**
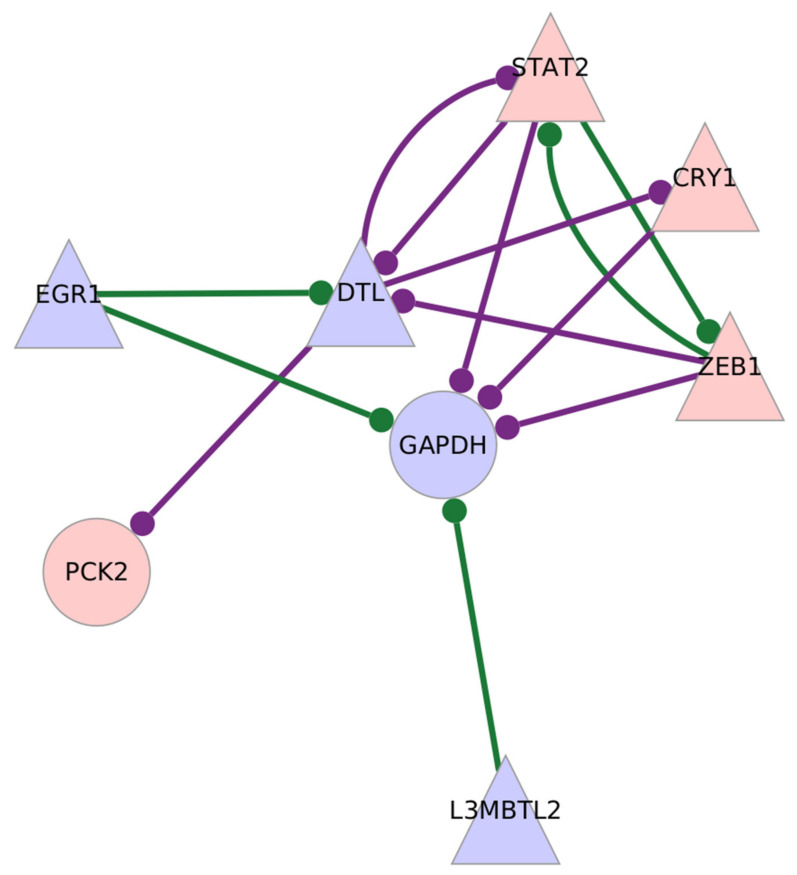
Subgraph of differentially expressed glycolysis-related enzymes with high centrality scores in the GTRD-annotated network of edges with significantly different edge weights between tocilizumab-treated RA and untreated RA samples, in CD8^+^ Tem cells. The differentially expressed transcription factors that link to the enzymes are shown. Triangles indicate transcription factors, and circles indicate target genes. Node colour shows the direction of differential expression: red nodes are over-expressed and blue nodes are under-expressed. The presence of an edge between a TF and a target gene refers to a significant difference in the edge weights of that gene pair between untreated RA samples and healthy samples. Purple edges show higher edge weight in RA, and green shows higher edge weight in healthy samples.

**Figure 8 genes-13-01216-f008:**
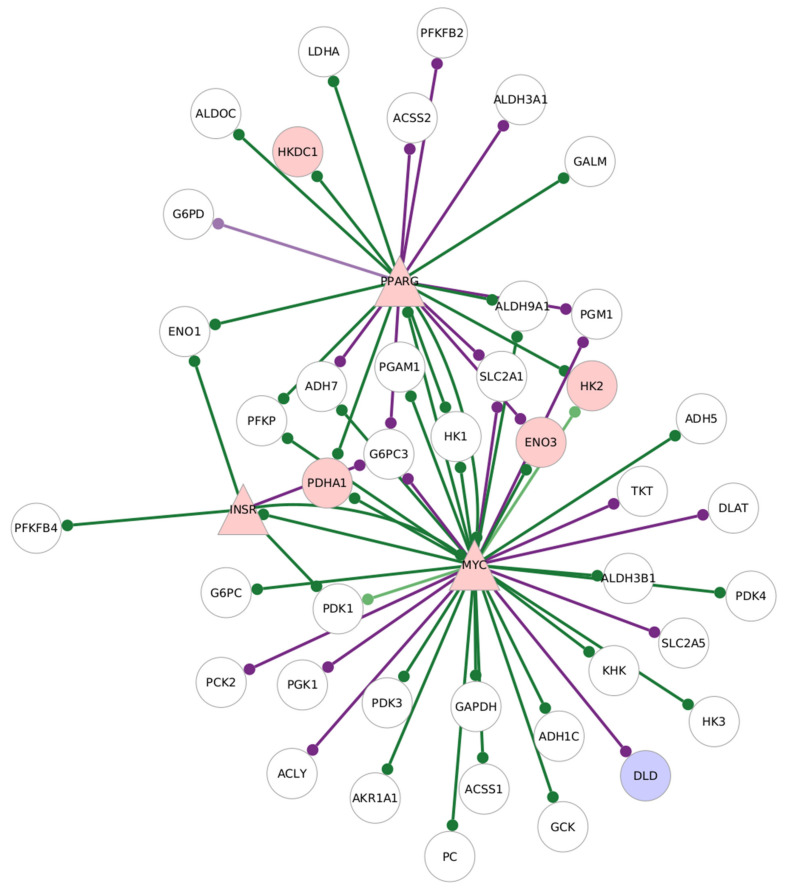
Subgraph of *MYC*, *PPARG* and *INSR* and their target genes in the GTRD-annotated network of edges with significantly different edge weights between untreated RA and healthy samples in CD8^+^ Temra cells. The differentially expressed transcription factors that link to the enzymes are shown. Triangles indicate transcription factors and circles indicate target genes. Node colour shows the direction of differential expression: red nodes are over-expressed, blue nodes are under-expressed and white nodes are not differentially expressed. The presence of an edge between a TF and a target gene refers to a significant difference in the edge weights of that gene pair between untreated RA samples and healthy samples. Purple edges show higher edge weight in RA, and green shows a higher edge weight in healthy samples.

**Figure 9 genes-13-01216-f009:**
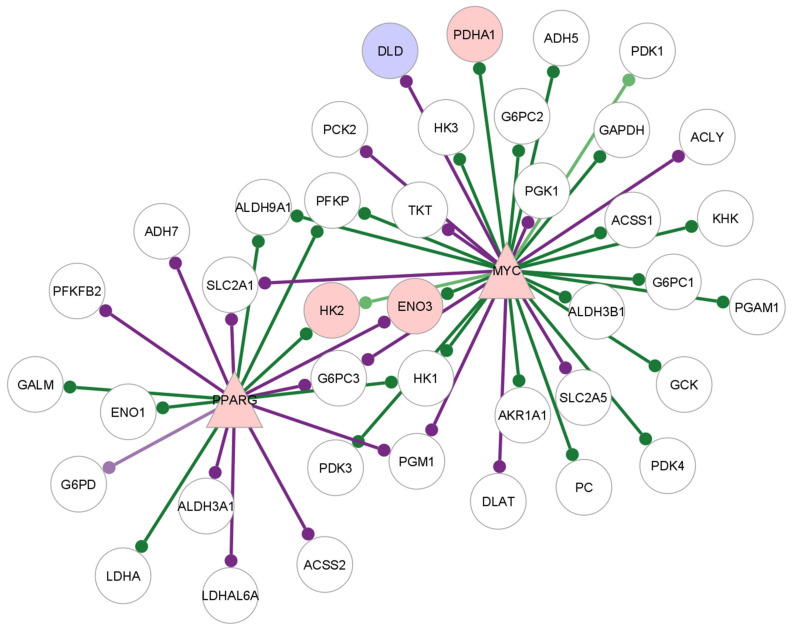
Subgraph of *MYC*, *PPARG* and *INSR* and their target genes in the GeneHancer annotated network of edges with significantly different edge weights between untreated RA and healthy samples, in CD8^+^ Temra cells. The differentially expressed transcription factors that link to the enzymes are shown. Triangles indicate transcription factors and circles indicate target genes. Node color shows the direction of differential expression: red nodes are over-expressed, blue nodes are under-expressed and white nodes are not differentially expressed. The presence of an edge between a TF and a target gene refers to a significant difference in the edge weights of that gene pair between untreated RA samples and healthy samples. Purple edges show higher edge weight in RA, and green shows higher edge weight in healthy samples.

**Figure 10 genes-13-01216-f010:**
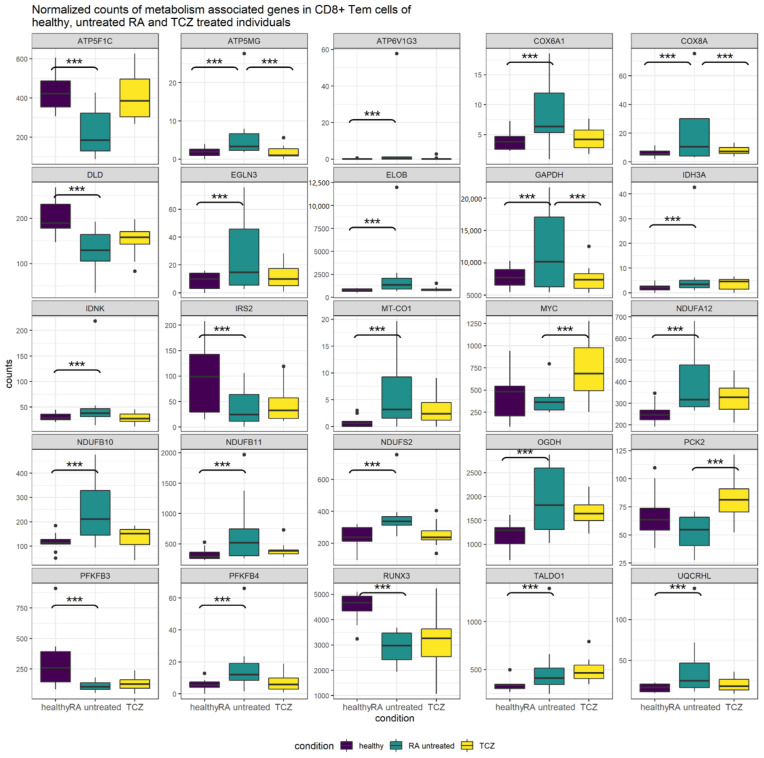
Normalized counts of genes associated with metabolic pathways or the regulation of glycolysis that show differential regulation in untreated RA CD8^+^ Tem cells or in TCZ-treated RA CD8^+^ Tem cells. Significant differential regulation is shown by asterisks (***).

**Figure 11 genes-13-01216-f011:**
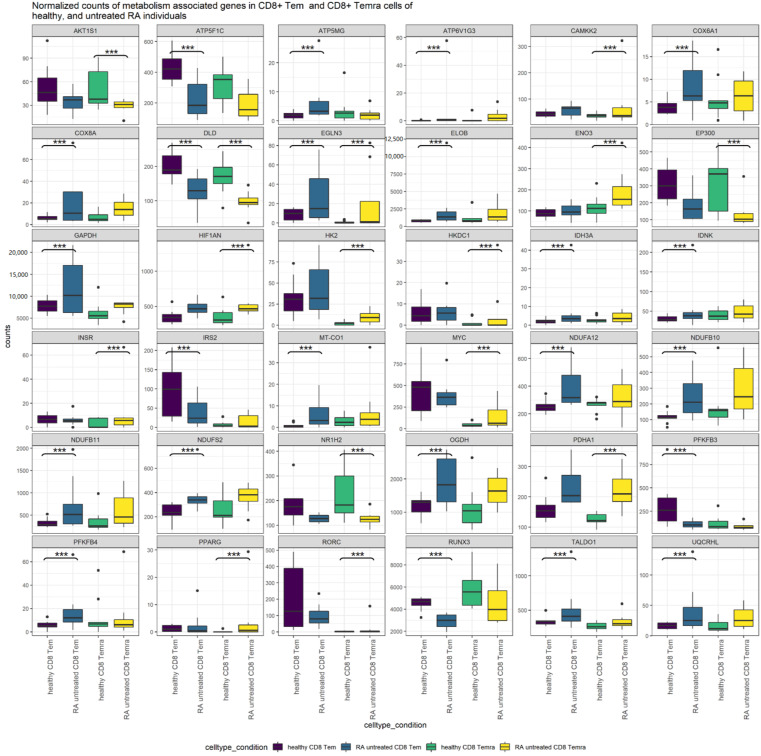
Normalized counts of genes associated with metabolic pathways or the regulation of glycolysis that show differential regulation in untreated RA CD8^+^ Tem cells or in untreated RA CD8^+^ Temra cells. Significant differential regulation is shown by asterisks (***).

**Figure 12 genes-13-01216-f012:**
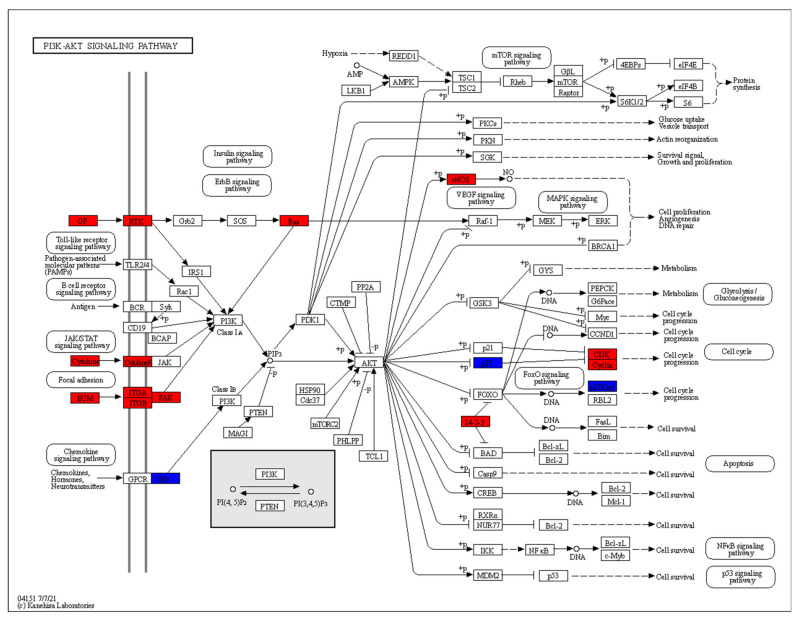
The differentially expressed genes of RA CD8^+^ Tem cells in the PI3K-AKT pathway. Red indicates upregulated genes and blue indicates downregulated genes.

**Figure 13 genes-13-01216-f013:**
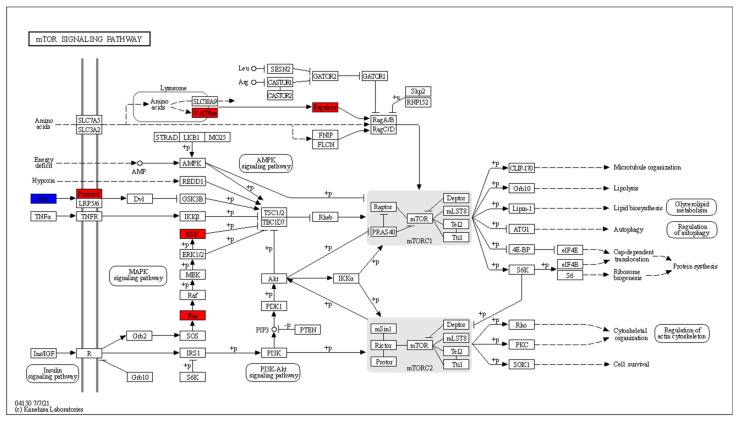
The differentially expressed genes of RA CD8^+^ Tem cells in the mTOR signalling pathway. Red indicates upregulated genes and blue indicates downregulated genes.

**Figure 14 genes-13-01216-f014:**
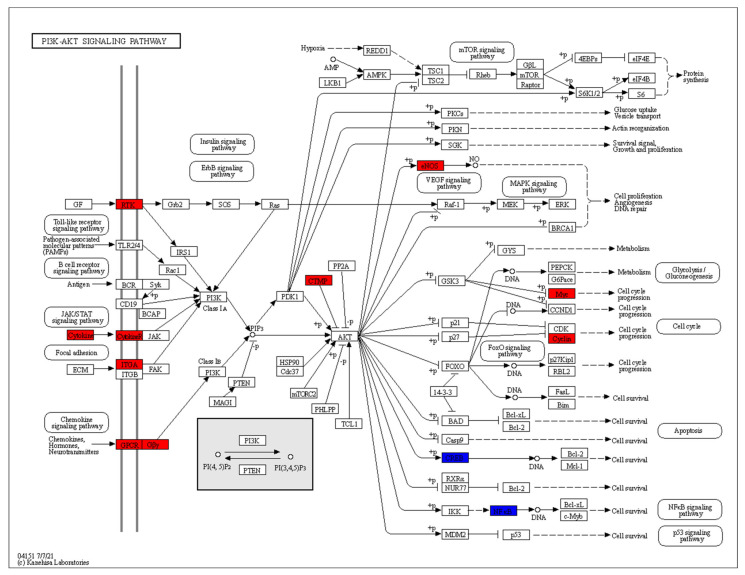
The differentially expressed genes of RA CD8^+^ Temra cells in the PI3K-AKT pathway. Red indicates upregulated genes and blue indicates downregulated genes.

**Figure 15 genes-13-01216-f015:**
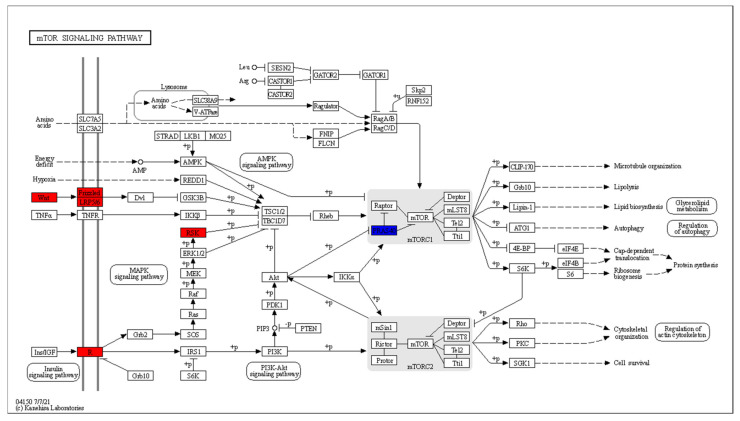
The differentially expressed genes of RA CD8^+^ Temra cells in the mTOR pathway. Red indicates upregulated genes and blue indicates downregulated genes.

**Figure 16 genes-13-01216-f016:**
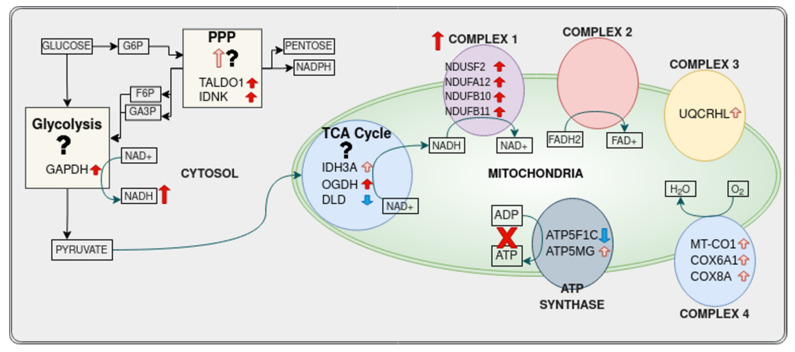
Differential expression of enzymes involved in glycolysis, pentose phosphate pathway, TCA cycle and oxidative phosphorylation in RA CD8^+^ Tem cells. Solid red UP arrows indicate upregulation/increased activity. Solid blue down arrows indicate downregulation. Light red arrows indicate statistically significant upregulation with low read counts. Pathways where the overall effect of differential regulation on the activity of the pathway is ambiguous are marked by a question mark. PPP—pentose phosphate pathway; G6P—glucose-6-phosphate; F6P—fructose-6-phosphate; GA3P—glyceraldehyde-3-phosphate.

**Figure 17 genes-13-01216-f017:**
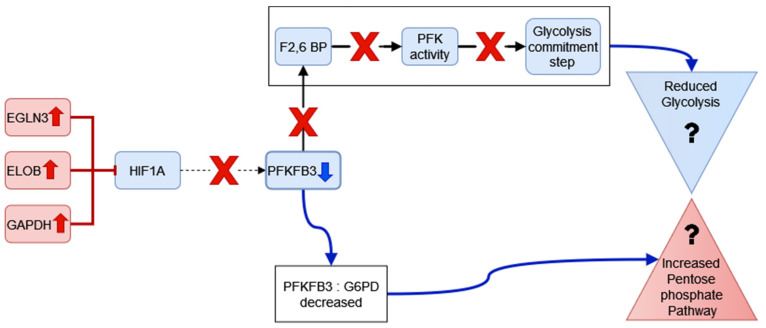
Proposed model for the regulation of glycolysis and pentose phosphate pathway in RA CD8^+^ Tem cells. *EGLN3*, *ELOB* and *GAPDH* are upregulated as indicated by the red up arrows. This leads to the inhibition of HIF1A (red T—shaped connector). HIF1A-mediated upregulation of *PFKFB3* (black dotted arrow) is blocked (red X) leading to the downregulation of *PFKFB3* (blue down arrow). The PFKFB3 catalysed increase in fructose 2, 6 bisphosphate (F 2,6 BP) is abolished, so F 2,6 BP-mediated allosteric activation of phosphofructokinase (PFK) is lost, and the commitment step of glycolysis does not occur. This leads to reduced glycolysis. *PFKFB3* downregulation causes a reduction in the ratio of PFKFB3 with G6PD (glucose 6 phosphate dehydrogenase) leading to increased pentose phosphate pathway activity.

**Figure 18 genes-13-01216-f018:**
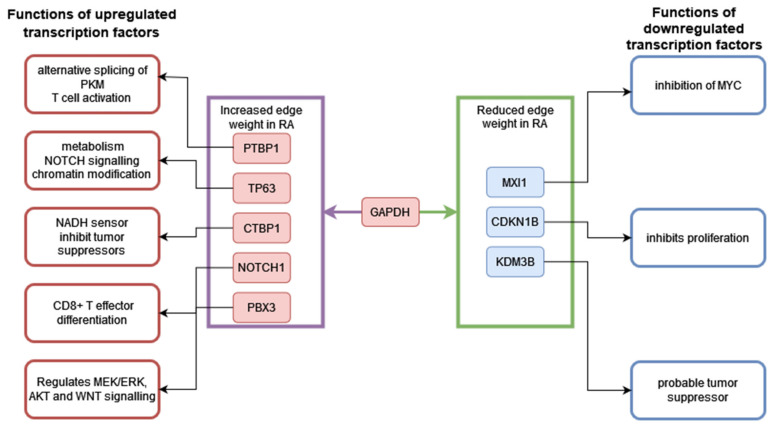
Differentially expressed transcription factors which show differential co-expression with GAPDH in RA CD8^+^ Tem cells. Red boxes show genes that are upregulated, and blue boxes show those that are downregulated. The transcription factors with increased edge weights are shown in the purple box and those with reduced edge weights with GAPDH in RA are shown in the green box. The functions of each transcription factor are also shown.

**Figure 19 genes-13-01216-f019:**
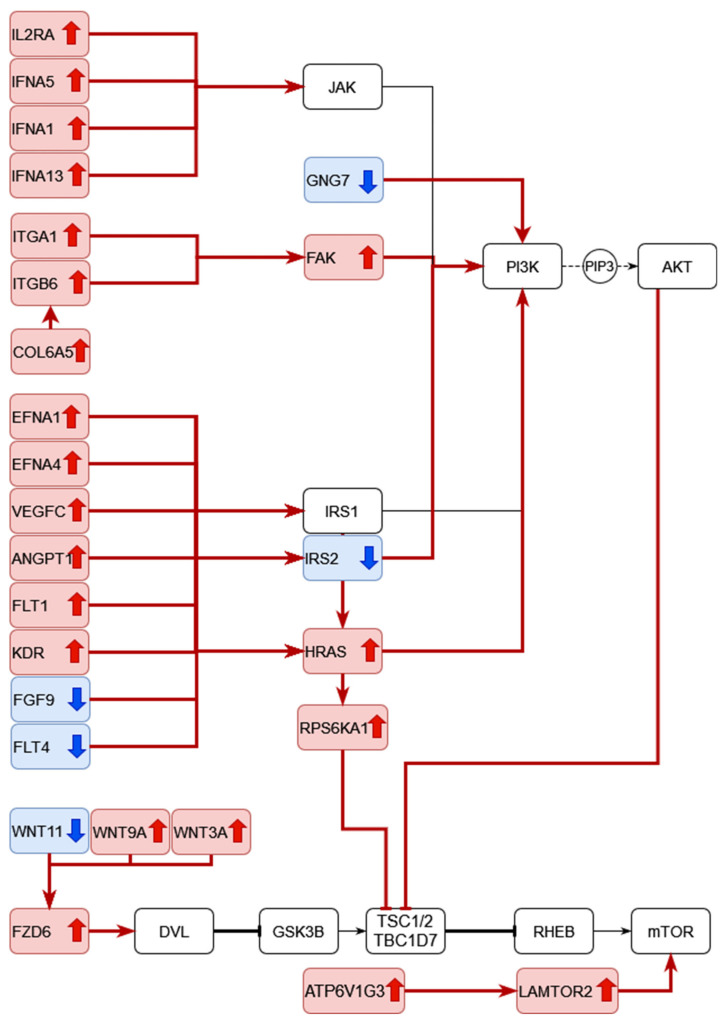
RA CD8^+^ Tem differentially expressed genes in the PI3K-AKT pathway and the mTOR pathway. Red up arrows indicate upregulation and blue down arrows indicate downregulation. Red connectors indicate the effect of the differentially regulated genes on their targets, whereas T shaped arrows indicate the inhibition and normal arrows indicate activation.

**Figure 20 genes-13-01216-f020:**
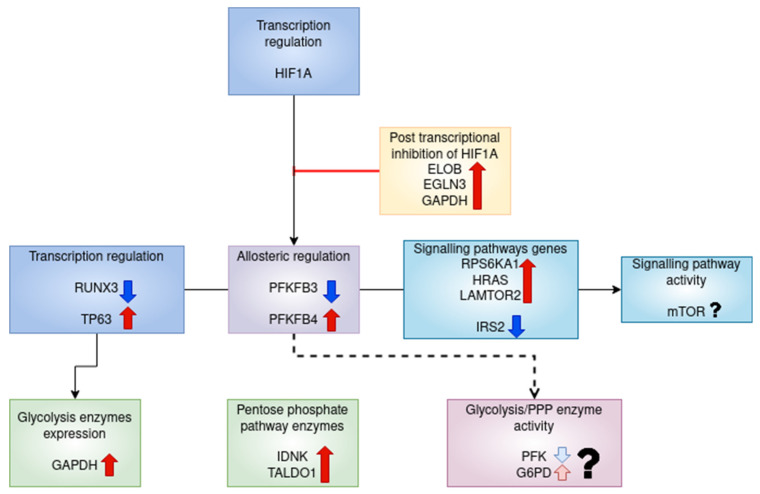
The differentially regulated genes involved in the regulation of glycolysis in RA CD8^+^ Tem cells. Post transcriptional inhibition of HIF1A by the upregulated *ELOB*, *EGLN3* and *GAPDH* (red up arrow) inhibits HIF1A activity resulting in the downregulation of *PFKFB3* (blue down arrow). This causes reduced activity of the glycolytic enzyme PFK (light blue down arrow) and increased activity of the pentose phosphate pathway enzyme G6PD (light red up arrow). The upregulated transcription factor *TP63* (red up arrow) causes increased expression of *GAPDH* (red up arrow). *IRS2* is downregulated (blue down arrow), while *RPS6KA1*, *HRAS* and *LAMTOR2* are upregulated (red up arrow) upstream of mTOR. The effect of these changes in mTOR cannot be determined in this study (question mark). The pentose phosphate pathway genes, *IDNK* and *TALDO1*, are also upregulated (red up arrow).

**Figure 21 genes-13-01216-f021:**
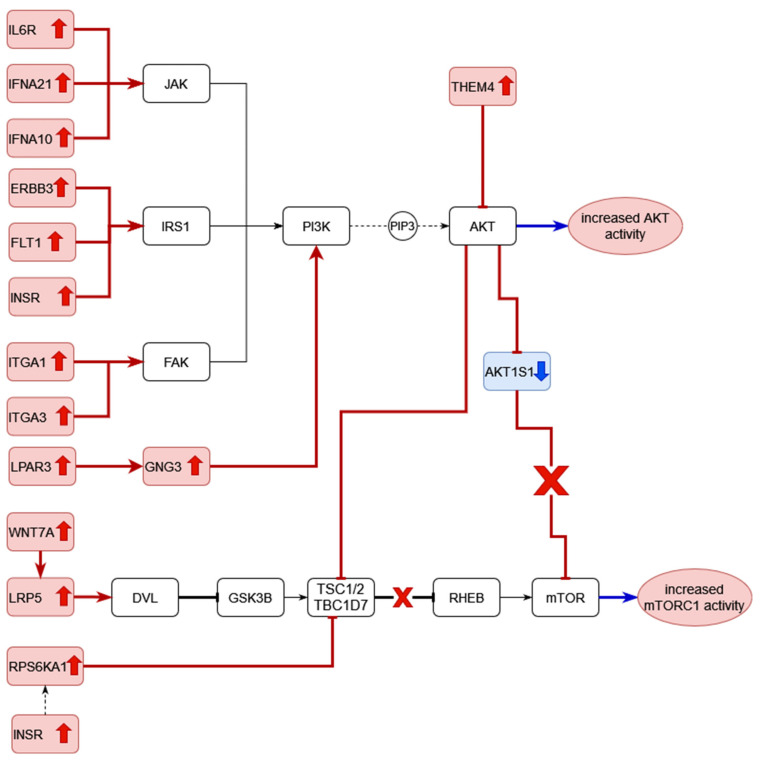
RA CD8^+^ Temra differentially expressed genes in the PI3K-AKT pathway and the mTOR pathway. Red up arrows indicate upregulation. Red arrows indicate the effect of the differentially regulated genes on their targets, where T shaped arrows indicate inhibition and normal arrows indicate activation. Dotted arrows indicate the presence of intermediate steps between the two connected proteins. Red crosses show the interactions that are lost in the RA CD8^+^ Temra cells due to inhibition or downregulation of the upstream proteins.

**Figure 22 genes-13-01216-f022:**
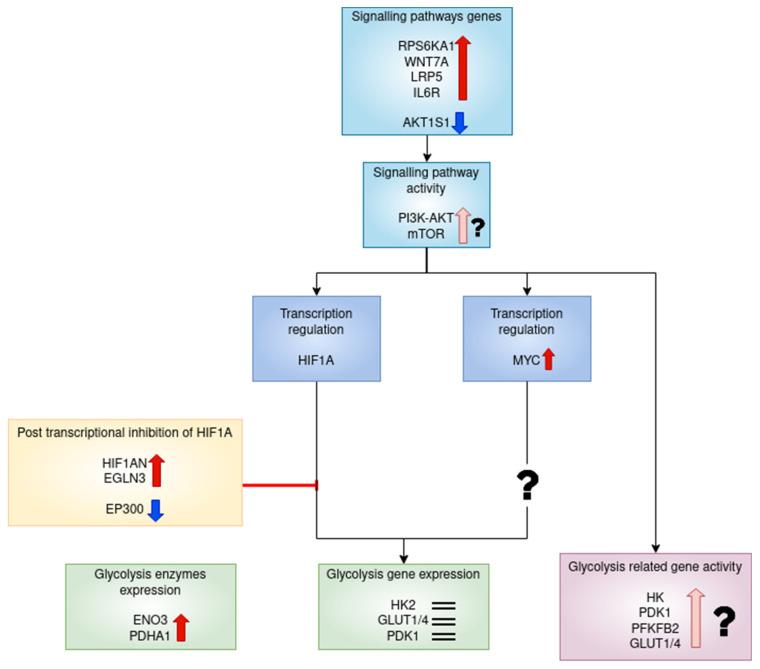
The differentially regulated genes involved in the regulation of glycolysis in RA CD8^+^ Temra cells. Post transcriptional inhibition of HIF1A by the upregulated *ELOB* and *EGLN3* (red up arrow) and the downregulation of HIF1A co-activator *EP300* (blue down arrow) inhibit HIF1A activity. *AKT1S1* is downregulated (blue down arrow), while *RPS6KA1*, *IL6R*, *WNT7A* and *LRP5* are upregulated (red up arrow) upstream of mTOR. AKT and mTOR may be activated in RA CD8^+^ Temra cells (light red up arrow and question mark). This may be the cause of the upregulation of *MYC* (red up arrow). Glycolysis target genes of *MYC* and *HIF1A* do not show differential regulation (black = signs). AKT and mTOR activity may increase the activity of glycolysis genes *HK*, *PDK1*, *PFKFB2* and GLUT1-4 (light red up arrow and question mark). *ENO3* and *PDHA1* are upregulated (red up arrow).

**Table 1 genes-13-01216-t001:** Details of the samples in the data set GSE118829.

Group	Cell Type	Total Samples	Gender	Age
Healthy control	CD4^+^ Tn	9	7f, 2m	45–77
	CD4^+^ Tcm	10	8f, 2m	
	CD4^+^ Tem	10	8f, 2m	
	CD8^+^ Tn	10	8f, 2m	
	CD8^+^ Tcm	8	6f, 2m	
	CD8^+^ Tem	10	8f, 2m	
	CD8^+^ Temra	9	7f, 2m	
Untreated RA	CD4^+^ Tn	10	8f, 2m	46–68
	CD4^+^ Tcm	10	8f, 2m	
	CD4^+^ Tem	9	7f, 2m	
	CD8^+^ Tn	9	7f, 2m	
	CD8^+^ Tcm	7	5f, 2m	
	CD8^+^ Tem	8	7f, 1m	
	CD8^+^ Temra	8	6f, 2m	
Infliximab-treated	CD4^+^ Tn	9	8f, 1m	44–66
	CD4^+^ Tcm	10	9f, 1m	
	CD4^+^ Tem	9	8f, 1m	
	CD8^+^ Tn	9	8f, 1m	
	CD8^+^ Tcm	10	9f, 1m	
	CD8^+^ Tem	6	5f, 1m	
	CD8^+^ Temra	8	7f, 1m	
Methotrexate-treated	CD4^+^ Tn	10	9f, 1m	28–76
	CD4^+^ Tcm	9	8f, 1m	
	CD4^+^ Tem	8	7f, 1m	
	CD8^+^ Tn	8	7f, 1m	
	CD8^+^ Tcm	4	3f, 1m	
	CD8^+^ Tem	9	8f, 1m	
	CD8^+^ Temra	6	5f, 1m	
Tocilizumab-treated	CD4^+^ Tn	10	10f	44–73
	CD4^+^ Tcm	10	10f	
	CD4^+^ Tem	10	10f	
	CD8^+^ Tn	10	10f	
	CD8^+^ Tcm	8	8f	
	CD8^+^ Tem	10	10f	
	CD8^+^ Temra	7	7f	
Synovial fluid	CD4^+^ Tn	0		71
	CD4^+^ Tcm	4	3f, 1m	
	CD4^+^ Tem	3	3f	
	CD8^+^ Tn	0		
	CD8^+^ Tcm	1	1m	
	CD8^+^ Tem	1	1f	
	CD8^+^ Temra	1	1f	

**Table 2 genes-13-01216-t002:** Upregulated and downregulated genes of untreated RA compared with healthy samples of T cell subsets.

Cell Type	Upregulated Genes	Downregulated Genes
CD4^+^ Tn	12	5
CD4^+^ Tcm	10	63
CD4^+^ Tem	3	1
CD8^+^ Tn	21	5
CD8^+^ Tcm	3	7
CD8^+^ Tem	1617	164
CD8^+^ Temra	843	291

**Table 3 genes-13-01216-t003:** Upregulated and downregulated genes of treated RA compared with untreated RA samples of T cell subsets.

Cell Type	Methotrexate		Infliximab		Tocilizumab	
	UP	DOWN	UP	DOWN	UP	DOWN
CD4^+^ Tn	1	9	12	4	4	20
CD4^+^ Tcm	1	1	1	8	2	4
CD4^+^ Tem	2	2	0	3	4	6
CD8^+^ Tn	16	51	0	26	19	711
CD8^+^ Tcm	3	6	2	3	3	3
CD8^+^ Tem	26	13	4	7	91	1047
CD8^+^ Temra	45	266	35	130	12	59

**Table 4 genes-13-01216-t004:** Enriched transcription factor of the downregulated gene list from untreated RA CD8^+^ Tem cells.

Term	Overlap	*p*-Value	Adjusted *p*-Value	Odds Ratio
CREB1 CHEA	30/1444	2.04 × 10^−6^	0.0001	2.92
UBTF ENCODE	33/1631	1.02 × 10^−6^	0.0001	2.88
CREB1 ENCODE	38/2238	9.19 × 10^−6^	0.0003	2.42
ZNF384 ENCODE	17/730	0.00011	0.0026	3.10
SUZ12 CHEA	25/1684	0.0027	0.0525	1.97
PBX3 ENCODE	20/1269	0.0038	0.0616	2.07
ATF2 ENCODE	36/2852	0.005	0.0689	1.69
BHLHE40 ENCODE	8/348	0.0081	0.0996	2.94
ZEB1 ENCODE	4/106	0.011	0.1010	4.84
MAX ENCODE	27/2073	0.010	0.1010	1.71

**Table 5 genes-13-01216-t005:** Enriched transcription factor of the downregulated gene list from untreated RA CD8^+^ Temra cells.

Term	Overlap	*p*-Value	Adjusted *p*-Value	Odds Ratio
TAF1 ENCODE	95/3346	1.69 × 10^−11^	1.73 × 10^−9^	2.45
CREB1 CHEA	51/1444	2.89 × 10^−9^	1.48 × 10^−7^	2.79
GABPA ENCODE	64/2082	4.99 × 10^−9^	1.69 × 10^−7^	2.47
UBTF ENCODE	53/1631	2.30 × 10^−8^	5.87 × 10^−7^	2.56
CREB1 ENCODE	65/2238	3.34 × 10^−8^	6.82 × 10^−7^	2.32
BRCA1 ENCODE	82/3218	1.14 × 10^−7^	1.95 × 10^−6^	2.073
ELF1 ENCODE	68/2483	1.45 × 10^−7^	2.11 × 10^−6^	2.18
ZNF384 ENCODE	30/730	3.34 × 10^−7^	4.25 × 10^−6^	3.12
ATF2 ENCODE	73/2852	6.29 × 10^−7^	7.13 × 10^−6^	2.04
YY1 ENCODE	70/2753	1.48 × 10^−6^	1.51 × 10^−5^	2.01

**Table 6 genes-13-01216-t006:** Glycolysis/gluconeogenesis pathway DEGs in untreated RA T cell subsets. Means are calculated from DESeq2 normalized counts.

Cell Type	Gene	adj. Pval	Log2FC	Linear FC	RA Mean	Control Mean
CD8^+^ Tem	*DLD*	0.071	−0.664	0.63	126.84	203.29
CD8^+^ Tem	*GAPDH*	0.052	0.631	1.55	12,014.37	7766.11
CD8^+^ Temra	*DLD*	0.046	−0.802	0.57	96.47	169.54
CD8^+^ Temra	*ENO3*	0.062	0.743	1.67	195.64	117.34
CD8^+^ Temra	*PDHA1*	0.00016	0.812	1.76	220.72	125.34
CD8^+^ Temra	*HKDC1*	0.017	2.414	5.33	6.095	1.14
CD8^+^ Temra	*HK2*	0.004	1.894	3.72	9.49	2.58

**Table 7 genes-13-01216-t007:** Glycolysis/gluconeogenesis pathway DEGs in MTX-, IFX- and TCZ-treated RA T cell subsets. Means are calculated from DESeq2 normalized counts.

Treatment	Cell Type	Gene	adj. Pval	Log2FC	Linear FC	Treatment Mean	RA Mean
TCZ	CD4^+^ Tn	*LDHA*	0.0008	−0.74	0.597	572.38	947.52
TCZ	CD8^+^ Tem	*PCK2*	0.097	0.67	1.56	82.23	52.53
TCZ	CD8^+^ Tem	*GAPDH*	0.055	−0.66	0.63	7657.78	12,014.37
MTX	CD8^+^ Temra	*HK2*	0.003	−3.01	0.12	1.23	9.49

**Table 8 genes-13-01216-t008:** Tricarboxylic acid cycle DEGs in untreated RA T cell subsets. Means are calculated from DESeq2 normalized counts.

Cell Type	Gene	adj. Pval	Log2FC	Linear FC	RA Mean	Control Mean
CD8^+^ Tem	*DLD*	0.07	−0.66	0.63	126.84	203.29
CD8^+^ Tem	*OGDH*	0.088	0.67	1.59	1924.58	1212.56
CD8^+^ Tem	*IDH3A*	0.045	1.7	3.25	8.13	2.08
CD8^+^ Temra	*DLD*	0.046	−0.802	0.574	96.47	169.54
CD8^+^ Temra	*PDHA1*	0.00016	0.81	1.76	220.72	125.34

**Table 9 genes-13-01216-t009:** Oxidative phosphorylation DEGs in untreated RA CD8^+^ Tem cells. Means are calculated from DESeq2 normalized counts.

Gene	adj. Pval	Log2FC	Linear FC	RA Mean	Control Mean
*COX6A1*	0.095	1.068	2.097	8.67	3.86
*NDUFB10*	0.074	1.076	2.109	247.30	115.93
*NDUFB11*	0.054	1.126	2.182	721.67	327.37
*ATP6V1G3*	0.051	5.091	34.072	7.59	0.06
*NDUFS2*	0.077	0.691	1.615	380.8	234.79
*ATP5F1C*	0.093	−0.89	0.539	225.44	425.32
*ATP5MG*	0.05	1.597	3.026	6.82	1.89
*COX8A*	0.018	1.685	3.214	20.92	6.11
*NDUFA12*	0.042	0.602	1.518	393.41	256.95
*MT-CO1*	0.036	2.88	7.36	6.4	0.86
*UQCRHL*	0.017	1.331	2.517	42.81	16.45

**Table 10 genes-13-01216-t010:** Oxidative phosphorylation DEGs in TCZ-treated RA T cell subsets. Means are calculated from DESeq2 normalized counts.

Cell Type	Gene	adj. Pval	Log2FC	Linear FC	Treatment Mean	RA Mean
CD8^+^ Tem	*ATP5MG*	0.059	−1.652	0.318	1.77	6.82
CD8^+^ Tem	*COX8A*	0.088	−1.334	0.397	8.03	20.93
CD8^+^ Tn	*NDUFB6*	0.088	−1.541	0.344	1.94	5.88
CD8^+^ Tn	*NDUFB8*	0.005	−2.297	0.203	3.2	17.97
CD8^+^ Tn	*MT-CYB*	0.066	−3.058	0.120	0.92	8.42
CD8^+^ Tn	*MT-ND5*	0.067	−2.555	0.170	2.48	14.81
CD8^+^ Tn	*MT-CO1*	0.071	−2.757	0.148	1.00	7.01

**Table 11 genes-13-01216-t011:** Fatty acid metabolism DEGs in untreated RA T cell subsets. Means are calculated from DESeq2 normalized counts.

Cell Type	Gene	adj. Pval	Log2FC	Linear FC	RA Mean	Control Mean
CD8^+^ Tem	*ACAA2*	0.099	3.165	8.972	6.4	0.65
CD8^+^ Temra	*PECR*	0.028	0.666	1.587	85.27	53.35
CD8^+^ Temra	*ACOT1*	1.0196 × 10^−5^	2.355	5.115	35.82	6.09

**Table 12 genes-13-01216-t012:** Fatty acid metabolism DEGs in treated RA T cell subsets. Means are calculated from DESeq2 normalized counts.

Treatment	Cell Type	Gene	adj. Pval	Log2FC	Linear FC	Treatment Mean	RA Mean
IFX	CD8^+^ Temra	*ACOT1*	6.64 × 10^−5^	−2.444	0.184	5.66	35.82
TCZ	CD8^+^ Tem	*ACACB*	0.022	0.726	1.654	145.99	89.00
TCZ	CD8^+^ Temra	*ACOT1*	0.0014	−2.407	0.189	5.83	35.82
TCZ	CD8^+^ Temra	*ACOT2*	0.029	−0.832	0.562	54.80	101.98

**Table 13 genes-13-01216-t013:** Number of significant edges and GTRD-annotated significant edges in T cell subsets.

Cell Type	RA Untreated vs. Healthy		IFX Treated vs. RA Untreated		MTX Treated vs. RA Untreated		TCZ Treated vs. RA Untreated	
	Total No: Significant Edges	Significant Edges Present in GTRD	Total No: Significant Edges	Significant Edges Present in GTRD	Total No: Significant Edges	Significant Edges Present in GTRD	Total No: Significant Edges	Significant Edges Present in GTRD
CD4^+^ Tn	185	127	2704	1160	41	24	404	196
CD4^+^ Tcm	15	5	0	0	1	0	753	257
CD4^+^ Tem	2039	1244	2521	1767	1551	978	6308	4205
CD8^+^ Tn	22,834	12,329	11,596	6475	19,241	12,321	15,036	8528
CD8^+^ Tcm	17,414	7874	331	234	1929	1470	38	32
CD8^+^ Tem	27,355	15,411	13,678	8197	28,653	15,138	17,151	9149
CD8^+^ Temra	128,318	75,606	96,729	53,945	84,906	48,879	82,597	45,443

**Table 14 genes-13-01216-t014:** Number of nodes and edges in the GTRD-annotated transcription factor–target gene network of untreated RA and treated RA T cell subsets.

Cell Type	RA Untreated vs. Healthy		IFX Treated vs. RA Untreated		MTX Treated vs. RA Untreated		TCZ Treated vs. RA Untreated	
	Total Nodes, TFs, Target Genes	Total Edges	Total Nodes, TFs, Target Genes	Total Edges	Total Nodes, TFs, Target Genes	Total Edges	Total Nodes, TFs, Target Genes	Total Edges
CD4^+^ Tn	92, 88, 4	127	650, 624, 26	1160	23, 23, 0	24	131, 123, 8	196
CD4^+^ Tcm	7, 7, 0	5	0	0	0	0	207, 202, 5	257
CD4^+^ Tem	627, 592, 35	1244	766, 728, 38	1767	588, 560, 28	978	1035, 984, 51	4205
CD8^+^ Tn	1173, 1115, 58	12,329	1074, 1020, 54	6475	1136, 1076, 60	12,321	1183, 1123, 60	8528
CD8^+^ Tcm	1072, 1018, 54	7874	182, 176, 6	234	438, 420, 18	1470	31, 31, 0	32
CD8^+^ Tem	1211, 1149, 62	15,411	1136, 1077, 59	8197	1248, 1183, 65	15,138	1146, 1092, 54	9149
CD8^+^ Temra	1285, 1214, 71	75,606	1279, 1211, 68	53,945	1281, 1211, 70	48,879	1284, 1215, 69	45,443

**Table 15 genes-13-01216-t015:** Top transcription factors and target genes in GTRD-annotated transcription factor–target gene network of untreated RA and treated RA T cell subsets.

Cell Type	RA Untreated v Healthy		IFX-treated v RA Untreated		MTX Treated v RA Untreated		TCZ Treated v RA Untreated	
	Top Ten TF	Top Ten Target Genes	Top Ten TF	Top Ten Target Genes	Top Ten TF	Top Ten Target Genes	Top Ten TF	Top Ten Target Genes
CD4^+^ Tn	-	-	*FOXA1*, *HOXA6*, *MITF*, *SMAD5*, *DNMT3B*, *RAD21*, *GATAD2B*	*PFKP*, *DLD*, *GAPDHS*, *PDK1*, *PGM2*, *ACSS1*	-	-	-	-
CD4^+^ Tcm	-	-	-	-	-	-	-	-
CD4^+^ Tem	*SPI1*, *PADI2*, *ZNF366*, *TP63*, *EHF*, *TWIST1*, *CBFA2T3*, *PRDM6*	*PDK4*, *GAPDH*, *PFKP*, *HK1*, *ALDH3B1*, *PGAM1*	*SPI1*, *PADI2*, *EHF*, *ZNF366*, *CBFA2T3*, *TP63*, *TWIST1*, *PPARG*, *VDR*	*ALDH3B2*, *PFKP*, *PGM2*, *ADPGK*, *ADH5*	*PADI2*, *ZNF366*, *SPI1*, *TP63*, *TWIST1*, *CBFA2T3*, *EHF*	*ENO3*, *PDK4*, *PGM1*, *GCK*	*PADI2*, *CBFA2T3*, *ETV1*, *SPI1*, *RBBP4*, *ZNF366*, *NR2F2*, *NFE2*	*ALDH3B2*, *ENO3*, *DLAT*, *PGM1*, *ACLY*
CD8^+^ Tn	*AHR*, *SALL3*, *FOXP1*, *KLF1*, *L3MBTL2*	*ALDH3B1*, *PCK1*, *ACSS2*, *KHK*, *ALDH7A1*, *PKM*, *ALDH3B2*, *AKR1A1*, *ENO3*	*MYOD1*, *AHR*, *RAD21*, *JUN*, *FOXP1*, *L3MBTL2*	*ALDH3B1*, *PCK1*, *ALDH7A1*, *KHK*, *ENO3*, *ACSS2*, *AKR1A1*, *GPI*	*NELFA*, *FOXP1*, *PHF8*, *HIF1A*, *RAD21*, *RING1*	*GPI*, *ALDH3B1*, *PCK1*, *PFKL*, *ADPGK*, *LDHA*, *ENO3*, *ACSS2*, *PFKFB3*	*TAL1*, *AHR*, *FOXP1*, *NR5A2*	*ALDH3B1*, *ALDH7A1*, *PCK1*, *KHK*, *ACSS2*, *PDK4*, *ENO3*, *AKR1A1*
CD8^+^ Tcm	*TEAD4*, *EGR2*, *SMC1A*, *FEZF1*	*PFKM*, *PDK2*, *ALDH3A1*, *ALDH3B1*, *PFKL*, *ALDH3A2*, *HK3*, *ADPGK*	-	-	-	-	-	-
CD8^+^ Tem	*MBL2*, *E2F8*, *ME1*, *ATF3*, *E2F7*	*PFKFB3*, *GAPDH*, *BPGM*, *DLD*, *G6PD*, *SLC2A1*, *PDHA1*, *PGAM1*	*E2F8*, *ME1*, *MBL2*, *E2F7*, *BRCA1*, *RBBP4*, *ATF3*, *GLIS1*, *CEBPD*	*GAPDH*, *ENO1*, *PGM1*, *PGK1*, *DLD*, *ENO3*, *PKM*	*MBL2*, *FOXP1*, *TCF7L1*, *EGR1*, *EGR2*, *ATF3*	*ENO2*, *ALDH2*, *GAPDH*, *ALDH9A1*, *ENO1*, *PGAM1*, *GCK*	*MBL2*, *E2F8*, *SMAD1*, *ME1*	*GAPDH*, *G6PD*, *ENO2*, *SLC2A1*, *G6PC*, *BPGM*, *PCK2*
CD8^+^ Temra	*MYC*, *ZNF143*, *TP53BP1*, *SP5*, *CDK7*	*ENO1*, *G6PC3*, *ALDOC*, *SLC2A1*, *ENO3*, *ACSS2*, *HK2*, *AKR1A1*, *HK1*	*ZNF143*, *TP53BP1*, *TEAD1*, *CREB1*	*ALDOC*, *ACSS2*, *ENO1*, *HK2*, *ENO3*, *G6PD*, *G6PC3*, *PFKFB2*, *AKR1A1*, *PGM2*	*RUNX1T1*, *MYC*, *ZNF143*, *INTS12*, *TEAD1*	*ALDOA*, *AKR1A1*, *ACSS2*, *ENO1*, *HK2*, *ALDOC*, *G6PC3*, *KHK*	*BRD3*, *TP53BP1*, *TEAD1*, *GRHL2*, *ZNF143*, *INTS12*	*ALDOA*, *ALDOC*, *G6PC3*, *ACSS2*, *ENO1*, *ENO3*, *HK2*, *PFKFB2*

**Table 16 genes-13-01216-t016:** Top ten target genes in GTRD- and GeneHancer-annotated transcription factor–target gene network of untreated RA CD8^+^ Tem cells.

Rank	Top Ten GTRD-Annotated Glycolysis Genes by In-Degree					Top Ten GeneHancer-Annotated Glycolysis Genes by In-Degree				
	Gene	DE	In-Degree in GTRD	In-Degree in GeneHancer	Common Edges *	Gene	DE	In-Degree in GTRD	In-Degree in GeneHancer	Common Edges *
1	*PFKFB3*	DOWN	103	79	53	*FBP2*	-	11	103	5
2	*GAPDH*	UP	60	36	21	*PFKFB3*	DOWN	103	79	53
3	*BPGM*	-	40	18	9	*DLD*	DOWN	39	36	20
4	*DLD*	DOWN	39	36	20	*GAPDH*	UP	60	36	21
5	*G6PD*	-	38	22	17	*ALDOB*	-	4	33	2
6	*SLC2A1*	-	31	24	14	*G6PC2*	-	3	24	1
7	*PDHA1*	-	25	13	10	*SLC2A1*	-	31	24	14
8	*PGAM1*	-	18	11	7	*G6PC*	-	12	22	9
9	*AKR1A1*	-	14	10	8	G6PD	-	38	22	17
10	*G6PC*	-	12	22	9	*HKDC1*	-	6	19	4

* Common edges refer to the number of TF—target edges for a particular glycolysis gene that is annotated by both databases.

**Table 17 genes-13-01216-t017:** Number of binding sites in GTRD and GeneHancer for *PFKFB3*, *DLD* and *GAPDH* in untreated RA CD8^+^ Tem cell differential co-expression networks.

Gene	Number of Common Transcription Factors in GeneHancer and GTRD	Number of TFs with More Binding Sites in GeneHancer	Number of TFs with More Binding Sites in GTRD	Number of TFs with the Same Number of Binding Sites in GTRD and GeneHancer
*PFKFB3*	53	48	2	3
*GAPDH*	21	16	4	1
*DLD*	20	12	4	4

**Table 18 genes-13-01216-t018:** Differential expression of regulators of glycolysis in untreated RA CD8^+^ Tem cells. Means are calculated from DESeq2 normalized counts.

Gene	adj. Pval	Log2FC	Linear FC	RA Mean	Control Mean
*IRS2*	0.072	−1.268	0.415	40.38	97.49
*RUNX3*	0.063	−0.617	0.652	2925.192	4511.98
*ELOB*	0.003	1.766	3.401	2695.83	785.48
*EGLN3*	0.073	1.616	3.064	27.06	8.55
*PFKFB3*	0.00097	−1.462	0.363	112.33	310.92
*PFKFB4*	0.042	1.611	3.054	18.22	5.88

CD8^+^ Tem cells from infliximab-treated individuals and methotrexate-treated individuals did not have any differentially expressed glycolysis-related genes when compared with untreated RA individuals. None of the transcription factors with high centrality scores were differentially expressed in either network.

**Table 19 genes-13-01216-t019:** Top ten target genes in GTRD- and GeneHancer-annotated transcription factor–target gene network of untreated RA CD8^+^ Tem cells.

Rank	Top Ten GTRD-Annotated Glycolysis Genes by In-Degree					Top Ten GeneHancer-Annotated Nodes by In-Degree				
	Gene	DE	In-Degree in GTRD	In-Degree in GeneHancer	Common Edges	Gene	DE	In-Degree in GTRD	In-Degree in GeneHancer	Common Edges
1	*ENO1*	-	189	135	88	*ENO1*	-	189	135	88
2	*G6PC3*	-	182	124	88	*G6PC3*	-	182	124	88
3	*ALDOC*	-	173	59	28	*G6PC2*	-	11	120	4
4	*SLC2A1*	-	155	118	77	*AKR1A1*	-	129	119	65
5	*ENO3*	UP	150	106	76	*SLC2A1*	-	155	118	77
6	*PDHA1*	UP	147	72	43	*ADH7*	-	60	109	17
7	*ACSS2*	-	146	104	71	*HK2*	UP	144	109	57
8	*HK2*	UP	144	109	57	*ENO3*	UP	150	106	76
9	*AKR1A1*	-	129	119	65	*HK1*	-	124	106	62
10	*HK1*	-	124	106	62	*ACSS2*	-	146	104	71

**Table 20 genes-13-01216-t020:** Expression levels of *AKR1A1* and *HK1* in untreated RA CD8^+^ Temra cells. Means are calculated from DESeq2 normalized counts.

Gene	adj. Pval	Log2FC	Linear FC	RA Mean	Control Mean
*AKR1A1*	0.05	0.56	1.48	92.08	60.7
*HK1*	0.203279	0.533165	1.46	414.55	288.62

**Table 21 genes-13-01216-t021:** The differentially expressed genes in RA CD8^+^ Temra cells that are involved in the regulation of glycolysis. Means are calculated from DESeq2 normalized counts.

Gene	adj. Pval	Log2FC	Linear FC	RA Mean	Control Mean
*INSR*	0.090	1.92	3.77	12.191	2.947
*PPARG*	0.010	4.47	22.09	4.497	0.136
*MYC*	0.00045	1.657	3.14	153.618	45.369
*AKT1S1*	0.086	−0.78	0.58	29.087	50.460
*EGLN3*	1.968 × 10^−5^	4.39	20.92	20.032	0.863
*HIF1AN*	0.00679	0.71	1.64	577.231	355.911
*EP300*	0.0346	1.12	2.18	137.036	301.163

## Data Availability

A publicly available dataset was analysed in this study. These data can be found here: GSE118829. The networks generated in this study are available as Appendix A.

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
