# Peer review of "Altered Transcriptional Regulation of Glycolysis in Circulating CD8+ T Cells of Rheumatoid Arthritis Patients"

_genes, 2022, doi:10.3390/genes13071216_

Round 1

Reviewer 1 Report

Harshan et al. have prepared a well-structured manuscript showing altered transcriptional regulation of glycolysis in RA patients by analyzing publicly available RNA sequencing data from circulating T lymphocyte subsets. They have also identified certain genes with high centrality scores and models for their differential expression in T lymphocyte subsets. This study has new networks generated which will be useful for the researchers in the same domain.
I would kindly request authors to add a paragraph delineating various drawbacks/limitations of the study to help other researchers. 

Reviewer 2 Report

In this manuscript, the authors conducted differential expressed gene and network analysis on public RNA-seq data from T lymphocyte subsets of healthy individuals and rheumatoid arthritis (RA) patients. They built differential co-expression networks and annotated the networks using the transcription factor - target gene interactions from the Gene Transcription Regulation Database (GTRD). Based on the networks, genes, and transcription factors that are related to T cells in RA or RA therapies were reported.

In general, the manuscript is in a good shape and I don't think it has serious flaws, overall. However, the authors need to improve this manuscript on the following issues.

Major comments:

1. The authors utilized the transcription factor target gene interactions from the GTRD database based on the presence of at least one binding site for a transcription factor in the region -1000 to +100 base pairs from the transcription start site of a given gene. The authors didn't state which cell types/ cell lines the TF binding data is from and whether it is related to RA T cells. The binding sites for the same TF in different cell types/tissues are also distinct so it makes no sense to assume that the TF binding profile in some other cell types can be used to infer the TF-target gene interactions in RA T cells. Also, it is too simplified to merely use the TF binding on genes' promoters to infer TF's target genes. Most TFs also bind on enhancers and can regulate the expression of target genes that are hundreds or even thousands of kilobasepair away. Identifying the target genes from a set of enhancers/TF binding sites is still an open question but the community has proposed methods such as ABC model(Fulco, Charles P., et al. "Activity-by-contact model of enhancer–promoter regulation from thousands of CRISPR perturbations." Nature genetics 51.12 (2019): 1664-1669.), RP model(Chen, Chen-Hao, et al. "Determinants of transcription factor regulatory range." Nature communications 11.1 (2020): 1-15.) to address this question. The authors need to define the link of TF-target genes in a more sophisticated way.

2. The authors used an FDR corrected p-value <= 0.1 and a linear fold change of 1.5 as cutoffs to select differentially expressed genes. Fold change at 1.5 is quite a stringent cutoff and whether the results are robust to the cutoff is unknown. The community has also developed many computational methods to infer the regulators using DE gene sets, such as Lisa(Qin et al. Genome biology(2020)) BART(Wang et al. Bioinformatics(2018)), i-cisTarget(Imrichova et al. Nucleic Acids Res(2015)), and Enrichr(Kuleshov et al. Nucleic Acids Res(2016)). It would be interesting to see what the regulators are for the DE genes called in this study.

3. The "results" section was full of descriptive statements. The authors should elaborate on which results are aligning with the previous literature and which are novel. 

Minor comments:

4. L276, "from the transcription start site of a given [10]". I think "gene" is missing here.

5. In the tables with the differentially expressed genes, the last columns were the mean expression level of the gene. The authors should state what is the number(TPM or FPKM? Raw values?)
